# CDK12 is hyperactivated and a synthetic-lethal target in BRAF-mutated melanoma

Thibault Houles [1], Geneviève Lavoie[1], Sami Nourreddine[1,7], Winnie Cheung[1], Éric Vaillancourt-Jean[1], Célia M. Guérin [1], Mathieu Bouttier[1], Benoit Grondin[1,8], Sichun Lin[2], Marc K. Saba-El-Leil[1], Stephane Angers [2,3,4], Sylvain Meloche [1,5] & Philippe P. Roux [1,6] ✉

Melanoma is the deadliest form of skin cancer and considered intrinsically resistant to chemotherapy. Nearly all melanomas harbor mutations that activate the RAS/mitogen-activated protein kinase (MAPK) pathway, which contributes to drug resistance via poorly described mechanisms. Herein we show that the RAS/MAPK pathway regulates the activity of cyclin-dependent kinase 12 (CDK12), which is a transcriptional CDK required for genomic stability. We find that melanoma cells harbor constitutively high CDK12 activity, and that its inhibition decreases the expression of long genes containing multiple exons, including many genes involved in DNA repair. Conversely, our results show that CDK12 inhibition promotes the expression of short genes with few exons, including many growth-promoting genes regulated by the AP-1 and NF-κB transcription factors. Inhibition of these pathways strongly synergize with CDK12 inhibitors to suppress melanoma growth, suggesting promising drug combinations for more effective melanoma treatment.

Melanoma is an aggressive malignancy originating from pigment-producing melanocytes localized at the epidermal-dermal junction in human skin[1]. While it is a rare type of skin cancer, melanoma accounts for a large majority of skin cancer-related deaths[2]. Early-stage primary melanomas are often cured by surgery[3], but up to 20% of patients go on to develop metastatic disease, which is one of the most lethal types of cancer[4]. The introduction of immunotherapies and targeted therapies has recently improved patient survival[5,6], but these therapies are limited by severe side effects, lack of clinical effects and rapidly emerging resistance mechanisms[7,8]. As such, only 23% of patients with metastatic melanoma survive five years (Melanoma Research Alliance, 2021).

Melanomas most often harbor mutations that affect the Ser/Thr kinase BRAF (50%), the small GTPase NRAS (25%), or the negative

regulator of RAS, neurofibromin 1 (NF1) (14%), resulting in increased RAS/mitogen-activated protein kinase (MAPK) signaling[4]. The RAS/MAPK pathway, which consists of RAS, RAF, MEK and ERK, plays many key roles in melanoma, making it an important therapeutic target[9]. The high rate of BRAF mutations has led to the generation of small-molecule inhibitors acting on mutated BRAF, including vemurafenib, which was the first FDA-approved BRAF inhibitor administered to advanced stage melanoma patients[10]. Despite early positive clinical results, patients were found to become resistant to BRAF inhibitors due to various mechanisms, including re-activation of ERK1/2 signaling by different means[11]. For this reason, the MEK1/2 inhibitor trametinib received FDA approval for combined administration with BRAF inhibitors[12]. However, patients receiving this treatment were shown to

[1]Institute for Research in Immunology and Cancer (IRIC), Université de Montréal, Montreal, 2950, Chemin de la Polytechnique, Montréal, QC H3T 1J4, Canada. [2]Donnelly Centre for Cellular & Biomolecular Research, Temerty Faculty of Medicine, University of Toronto, Toronto, ON, Canada. [3]Leslie Dan Faculty of Pharmacy, University of Toronto, Toronto, ON, Canada. [4]Department of Biochemistry, Temerty Faculty of Medicine, University of Toronto, Toronto, ON, Canada. [5]Department of Pharmacology and Physiology, Faculty of Medicine, Université de Montréal, Montreal, QC, Canada. [6]Department of Pathology and Cell Biology, Faculty of Medicine, Université de Montréal, Montreal, QC, Canada. [7]Present address: Department of Bioengineering, University of California, San Diego, San Diego, CA, USA. [8]Present address: Department of Biological Sciences, Université du Québec à Montréal, Montreal, QC, Canada. ✉e-mail: philippe.roux@umontreal.ca

become irresponsive within several months of treatment[7,8]. Constitutive activation of the RAS/MAPK pathway also promotes resistance to conventional chemotherapeutic drugs by modulating the DNA damage response[7,13,14], but the molecular mechanisms by which this occurs remain elusive.

CDK12 belongs to the cyclin-dependent kinase (CDK) family of Ser/Thr kinases and has recently been found to have multiple roles in gene expression and tumorigenesis[15]. CDK12 was originally shown to associate with cyclin K to phosphorylate RNA polymerase II (Pol II) within its C-terminal domain (CTD) heptad repeat ($Y_1S_2P_3T_4S_5P_6S_7$), which serves as a platform for the recruitment of factors controlling transcriptional and post-transcriptional events[16,17]. Pol II phosphorylation by CDK12 regulates transcription elongation, splicing, as well as cleavage and polyadenylation[18,19]. CDK12 also suppresses intronic polyadenylation[20,21], which partly explains why long genes with many exons are particularly sensitive to CDK12 inhibition, such as those involved in DNA replication, recombination and repair[16,22]. While CDK12 was shown to be required for proper execution of the DNA damage response[23], the mechanisms involved in CDK12 regulation remain unknown. Based on the crystal structure of the CDK12/cyclin K complex[24], potential regulatory mechanisms include T-loop phosphorylation by an upstream kinase and/or protein-protein interactions with potential binding partners[15].

In this study, we use a proteomics approach to identified CDK12 as a potential effector of the RAS/MAPK pathway. Our results show that ERK1/2 promote CDK12 phosphorylation and thereby positively regulate its kinase activity. Consistent with this, we found that CDK12 is constitutively activated in BRAF-mutated melanoma and that its inhibition impairs the expression of long genes with multiple exons, including genes involved in the DNA damage response. We also find that CDK12 inhibition promotes the expression of short genes with few exons, including many targets of the stress-responsive AP-1 and NF-κB pathways. Our results show that inhibition of these pathways strongly synergize with CDK12 inhibitors to inhibit melanoma growth, suggesting potential combinations for the treatment of melanoma patients.

## Results

### Characterization of the proximity interactome of ERK1 and ERK2

To characterize the global proximity partners of ERK1 and ERK2, we performed a proximity-dependent biotin identification (BioID) screen using mass spectrometry (MS). For this, HEK293 cells were transiently transfected with GFP, ERK1 or ERK2 fused to a promiscuous form (R118G) of the biotin ligase BirA (BirA*) (Supplementary Fig. 1a). ERK1/2 fusion proteins were found to retain their abilities to interact with and phosphorylate RSK1/2 (Supplementary Fig. 1b, c), which are bona fide ERK1/2 substrates[25]. The BioID screen was performed in biological triplicates originating from cells grown under standard conditions. High-confidence proximity interactions were revealed for ERK1 and ERK2, compared to GFP, using the Significance Analysis of INTeractome (SAINT) software package[26,27] (Fig. 1a). BirA*-ERK1, BirA*-ERK2 and BirA*-GFP were shown to be expressed at similar levels, and to result in robust protein biotinylation patterns (Fig. 1b). Using these cells, we found a very strong correlation ($r = 0.918$) between identified ERK1 and ERK2 proximity interactors (Fig. 1c; Supplementary Data 1), suggesting that ERK1 and ERK2 have very similar proximity interactomes. We thus combined results obtained for ERK1 and ERK2, which resulted in the identification of 179 high-confidence prey proteins (SAINT score ≥ 0.80, BFDR ≤ 0.05 and Log2 fold-change (FC) ≥ 1, as statistical thresholds) (Fig. 1d; Supplementary Data 1). Interestingly, we found that ERK1/2 proximity partners were mostly implicated in transcription ($p$-value = $2.83 \times 10^{-6}$), mRNA processing ($p$-value = $6.43 \times 10^{-16}$), DNA repair ($p$-value = $5.17 \times 10^{-5}$), cell trafficking ($p$-value = $2.83 \times 10^{-6}$), ribosome biogenesis ($p$-value = $2.35 \times 10^{-6}$),

cytoskeletal reorganization ($p$-value = $1.91 \times 10^{-3}$), and cell cycle regulation ($p$-value = $6.41 \times 10^{-5}$) (Fig. 1e).

To identify potential substrates of ERK1/2, we compared our results with proteins listed in the ERK1/2 compendium[28], which is a comprehensive list of proteins that have been shown to be direct or indirect substrates of the RAS/MAPK pathway. In addition, we used the bioinformatics tool Scansite[29] to identify proteins that contain a predicted ERK interaction domain (ERK binding domain or D domain) and/or a potential ERK1 kinase motif (using medium stringency as threshold). Through these analyses, we found that 72 potential ERK1/2 proximity interactors (40%) contained both a predicted ERK interaction domain and ERK1 kinase motif (Fig. 1f). Moreover, we found that 56 proximity interactors (31%) were either direct or indirect ERK1/2 substrates (Fig. 1f). To increase the likelihood of identifying direct ERK1/2 substrates, we looked at the intersection of all three lists and found 31 potential ERK1/2 substrates. From these proteins, 8 (26%) were previously shown to be direct ERK1/2 substrates, such as JUN[30], GAB1[31], STIM1[32] and EP300[33]. Globally, our results suggest the identification of 23 potential ERK1/2 substrates, including CDK12, which was selected for further characterization.

### ERK1/2 directly phosphorylate CDK12 on conserved Thr548

Based on our BioID analysis, we identified CDK12 as a potential ERK1/2 substrate (Fig. 1f). To verify this, we used a phospho-motif antibody that recognizes the pSer/Thr-Pro (pS/T-P) motif, which is typically phosphorylated by ERK1/2[34]. HEK293 cells were transfected with Myc-tagged human CDK12 and serum-starved overnight to lower basal ERK1/2 activity. Cells were then stimulated with phorbol 12-myristate 13-acetate (PMA) and epidermal growth factor (EGF), which are strong RAS/MAPK agonists, or insulin and serum to more strongly stimulate the PI3K/AKT pathway, for the indicated times (Fig. 2a). Immunoprecipitated CDK12 was then analyzed for phosphorylation by immunoblotting using the anti-pS/T-P antibody. We found that PMA and EGF, two potent agonists of the RAS/MAPK pathway (as shown by ERK1/2 phosphorylation), stimulated CDK12 phosphorylation in HEK293 cells. Interestingly, activation of the PI3K/AKT pathway (as shown by AKT phosphorylation) by insulin or serum did not significantly increase CDK12 phosphorylation, suggesting that CDK12 is mainly regulated by the RAS/MAPK pathway in HEK293 cells. This was further investigated using pharmacological inhibitors against ERK1/2 (BVD-523) and MEK1/2 (PD184352), which decreased CDK12 phosphorylation induced by PMA stimulation (Fig. 2b, c). CDK12 phosphorylation does not appear to affect its interaction with Cyclin K, its main binding partner (Fig. 2a–c). To confirm the direct involvement of ERK1/2, we performed in vitro kinase assays using purified fragments of CDK12 (Fig. 2d) and γ[32P]ATP. Recombinant active ERK1 was incubated in a kinase reaction buffer with bacterially-purified GST-tagged CDK12 fragments (F1, F2, F3, and F4) (Fig. 2d). We found that activated ERK1 robustly increased [32P] label incorporation into purified GST-CDK12 F2, but not into F1, F3, and F4 (Fig. 2e), suggesting that ERK1/2 predominantly phosphorylate CDK12 between aa 352 and 711. Using Scansite, we verified the presence of potential ERK1/2 phosphorylation sites, which predicted four potential sites in CDK12 (Thr525, Ser534, Thr548, Ser681) (Fig. 2d). In accordance to the in vitro phosphorylation results, all predicted sites in CDK12 are located in the proline-rich motif comprised within F2 (Fig. 2d). In order to identify the exact site(s) phosphorylated by ERK1/2, we generated non-phosphorylatable CDK12 mutants (T525A, S534A, T548A, S681A). With this approach, we found that mutation of Thr548 completely abrogated CDK12 phosphorylation induced by PMA stimulation (Fig. 2f). CDK12 sequences from different species were aligned to show the conservation of Thr548 across vertebrate species (Fig. 2g). Mutation of Thr548 also completely abrogated CDK12 phosphorylation induced by an activated form of MEK1 (MEK-DD; S218/222D) (Fig. 2h), suggesting that the RAS/MAPK pathway is sufficient for stimulating Thr548 phosphorylation. Lastly,

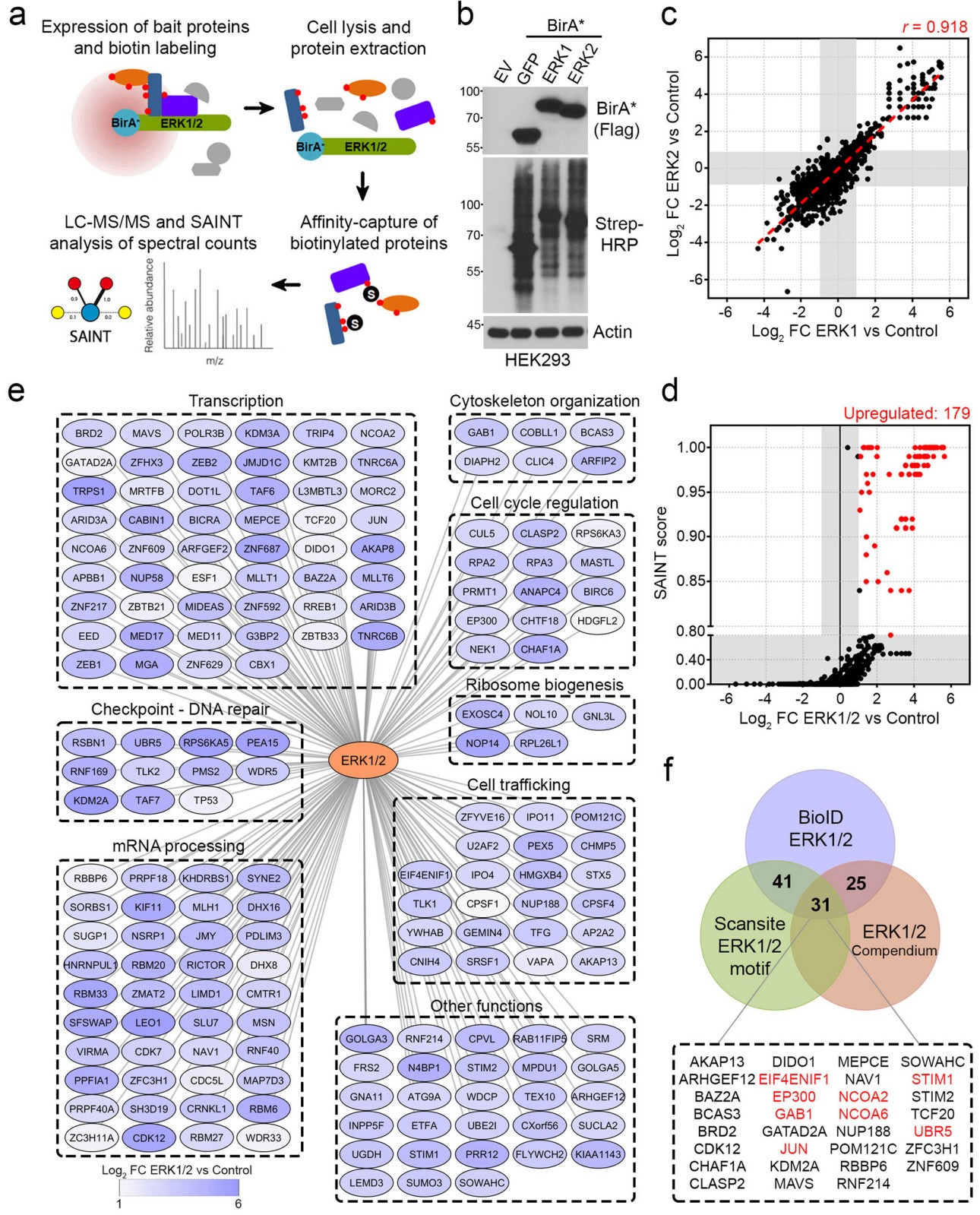

we generated a phospho-specific antibody directed against Thr548, and found that activation of the RAS/MAPK pathway correlated with the phosphorylation of ectopic CDK12 on this site (Fig. 2i). The use of wild-type (WT) CDK12, the T548A mutant and the treatment with λ-phosphatase confirmed the specificity of the phospho-specific antibody (Fig. 2i), and as expected, we found that inhibition of ERK1/2 activation (using PD185352 and BVD-523) reduced Thr548 phosphorylation after PMA stimulation (Fig. 2j). Together, these results strongly

suggest that CDK12 is directly phosphorylated by ERK1/2 on Thr548 in response to RAS/MAPK pathway activation.

## ERK1/2 promote CDK12 activation in BRAF-mutated melanoma cells

Based on DepMap data (https://depmap.org/), CDK12 is a common essential gene in cancer cell lines from different origins, including melanoma[35]. One of its functions in cells is to phosphorylate at least

**Fig. 1 | Characterization of the proximity interactome of ERK1 and ERK2.**
**a** Schematic representation of the proteomics approach used (BioID, proximity-dependent biotin identification) to characterize the proximity interactome of ERK1 and ERK2. **b** The expression of each transfected bait (BirA-GFP, BirA-ERK1 and BirA-ERK2) in HEK293 was assessed by immunoblotting against the Flag epitope. As cells were also labeled with biotin (50 μM, 24 h), global proximity biotinylation was evaluated by immunoblotting using Streptavidin-HRP. EV empty vector.
**c** Correlation plot between ERK1 and ERK2 proximity interactors identified by BioID. Shaded areas represent cut-off range of Log$_2$ FC (ERK1 versus Control, or ERK2 versus Control). The $r = 0.918$ value was determined using Pearson correlation. **d** Global ERK1/2 proximity interactors identified by BioID. Shaded areas

represent cut-off range of SAINT score and Log$_2$ FC (ERK1/2 versus Control).
**e** Network of the ERK1/2 proximity interactome organized by biological processes found to be significantly enriched. **f** Venn diagram representing the overlap between ERK1/2 proximity interactors identified by BioID (this study), and proteins containing a predicted ERK1 motif (kinase and binding site, Scansite 4.0) or present in the ERK1/2 target protein compendium (http://sys-bio.net/erk_targets/). Proteins listed in red are known ERK1/2 substrates. Data are from $n = 3$ (**b**, **c**) and $n = 6$ (**d**) independent experiments. **d–f** Only preys identified by BioID with a SAINT Score ≥ 0.8, BFDR ≤ 0.05 and Log$_2$ FC ≥ 1 were considered. Source data are provided as a Source Data file.

two serine residues (Ser2 and Ser5) within the CTD of the Pol II subunit POLR2A (RPB1), an essential step in transcription elongation[24]. To determine if CDK12 activity is regulated by the RAS/MAPK pathway, we developed an in vitro kinase assay based on endogenous CDK12. To validate the assay, CDK12 was immunoprecipitated from serum-growing HEK293 cells and incubated in a kinase reaction buffer with bacterially-purified GST or GST-RPB1, and unlabeled ATP. In vitro phosphotransferase activity was detected with phospho-specific antibodies targeting Ser2 and Ser5 of RPB1, whereas total RPB1 levels were measured as control. Using this assay, we found that endogenous CDK12 efficiently phosphorylates Ser2 and Ser5 of GST-RPB1, compared to GST alone (Supplementary Fig. 2a). We also found that treatment of CDK12 immunoprecipitates with THZ531, a potent CDK12 inhibitor[36], abrogated CDK12-mediated phosphorylation of Ser2 and Ser5 (Supplementary Fig. 2a). Using this validated approach, we found that PMA stimulation robustly increased endogenous CDK12 kinase activity in serum-starved HEK293 cells (~7 fold-change), which was prevented by pre-treatments with either MEK1/2 (PD185352) or ERK1/2 (BVD-523) inhibitors (Fig. 3a). These results indicate that endogenous CDK12 kinase activity correlates with the RAS/MAPK pathway, and suggest that ERK1/2 might directly participate in the regulation of CDK12 activity.

Melanomas are characterized by the hyperactivation of the RAS/MAPK pathway, and consistent with this, ERK1/2 activity is frequently elevated in cells derived from this type of cancer[12]. Compared to melanocytes, we found that several BRAF-mutated melanoma cell lines display high levels of RPB1 Ser2 and Ser5 phosphorylation (Supplementary Fig. 2b), which could be indicative a higher CDK12 activity. Consistent with this, we found that several BRAF-mutated melanoma cell lines are more sensitive to CDK12 inhibition (THZ531) (IC$_{50}$ values between 55 nM and 220 nM) than melanocytes (IC$_{50}$ > 1 μM) (Fig. 3b). To measure the kinase activity of CDK12, we performed in vitro kinase reactions using endogenous CDK12 immunoprecipitated from A375, a BRAF-mutated melanoma cell line. As expected, basal CDK12 kinase activity was found to be high in serum-starved A375 cells (Fig. 3c). Importantly, CDK12 activity was severely abrogated by treatments with MEK1/2 (PD184352) or ERK1/2 (BVD-523) inhibitors (Fig. 3c), suggesting that CDK12 activity is indeed regulated by the RAS/MAPK pathway in these cells. In complement to CDK12 kinase activity, we measured RPB1 phosphorylation and the expression of CDK12 target genes in BRAF-mutated melanoma cell lines. In addition to reducing RPB1 phosphorylation in A375 (Fig. 3d) and Colo829 (Supplementary Fig. 2c) cells, we found that THZ531 decreased the expression of genes involved in the DNA damage response (Supplementary Fig. 2d, e). Interestingly, MEK1/2 and ERK1/2 inhibition resulted in similar reductions in RPB1 phosphorylation (Fig. 3d; Supplementary Fig. 2c) and DDR gene expression (Supplementary Fig. 2f, g), consistent with the idea that CDK12 is regulated by the RAS/MAPK pathway.

To determine more precisely if ERK1/2-mediated CDK12 phosphorylation plays a role in its regulation, we first monitored Thr548 phosphorylation in A375 cells expressing Myc-tagged CDK12. These cells were found to display constitutively high levels of CDK12 phosphorylation at Thr548 (Fig. 3e), which was sensitive to both MEK1/2

(PD184352) and ERK1/2 (BVD-523) inhibitors. To determine if Thr548 phosphorylation participates in the regulation of CDK12 activity, wild-type (WT) CDK12 or the T548A mutant were immunoprecipitated from A375 cells to measure CDK12 activity in an in vitro kinase assay (Fig. 3f). These experiments revealed that mutation of Thr548 reduces CDK12 kinase activity compared to wild-type CDK12, suggesting that ERK1/2-mediated CDK12 phosphorylation contributes to its activation by the RAS/MAPK pathway.

## CDK12 inhibition increases the expression of short-growth-promoting genes

To determine the role of CDK12 in melanoma, we performed RNA-sequencing (RNA-seq) on two BRAF-mutated melanoma cell lines (A375 and Colo829) treated with CDK12 inhibitors (THZ531; 500 nM) for 6 h (Supplementary Data 2). Transcriptome analysis revealed a large number of downregulated (2333 in Colo829, and 1737 in A375) and upregulated (1158 in Colo829, and 835 in A375) genes (Fig. 4a), which were found to correlate to a high degree ($r = 0.708$) between both cell lines (Fig. 4b). We thus combined transcriptome data from A375 and Colo829 cells, which revealed a total of 1923 and 875 genes that were significantly downregulated and upregulated in response to CDK12 inhibition ($p$-value ≤ 0.05 and Log$_2$ fold-change (FC) ≥ 1), respectively (Fig. 4c). Based on the role of CDK12 in transcription elongation[37], we determined the length of genes affected by CDK12 inhibition and found a negative correlation between gene length and overall fold-change (slope = −0.3297) (Fig. 4d). Indeed, longer genes (≥14.6 kb) with multiple exons (≥ 8 exons on average) were more likely to be downregulated in response to CDK12 inhibition (Fig. 4e, f; Supplementary Fig. 3a). Based on Gene Set Enrichment Analysis (GSEA)[38] and qPCR experiments, we found that CDK12 inhibition significantly decreased transcripts associated with homologous recombination (HR) in both A375 and Colo829 cells (Fig. 4g; Supplementary Fig. 2d, e). Consistent with this, we found that increasing concentrations of THZ531 decreased BRCA1 and Rad51 protein level (Fig. 4h; Supplementary Fig. 3f), which correlates with a dose-dependent increase in DNA damage in both melanoma cell lines (Fig. 4i; Supplementary Fig. 3b).

Surprisingly, we also found that short genes (≤14.6 kb) with fewer exons (≤4 exons on average) were more likely to be upregulated in response to CDK12 inhibition (Fig. 4e, f; Supplementary Fig. 3a). GSEA analysis revealed that upregulated genes were enriched for the NF-κB (Fig. 4j) and AP-1 (Fig. 4k) pathways in both A375 and Colo829 cells. To validate the regulation of these pathways by CDK12, we treated both melanoma cell lines with increasing concentrations of two CDK12 inhibitors (THZ531 and SR-4835). Interestingly, CDK12 inhibition was found to increase gene and protein expression of several AP-1 and NF-κB targets in a dose-dependent manner (Fig. 4h, l; Supplementary Fig. 3c–f). In addition, we found that shRNA-mediated CDK12 knockdown resulted in similar changes in *JUN* and *FOS* expression, two transcripts coding for key AP-1 effectors (Supplementary Fig. 3g). Interestingly, CDK12 inhibition did not result in similar effects in melanocytes (Supplementary Fig. 3h, i). Collectively, these results demonstrate that CDK12 inhibition results in the specific upregulation

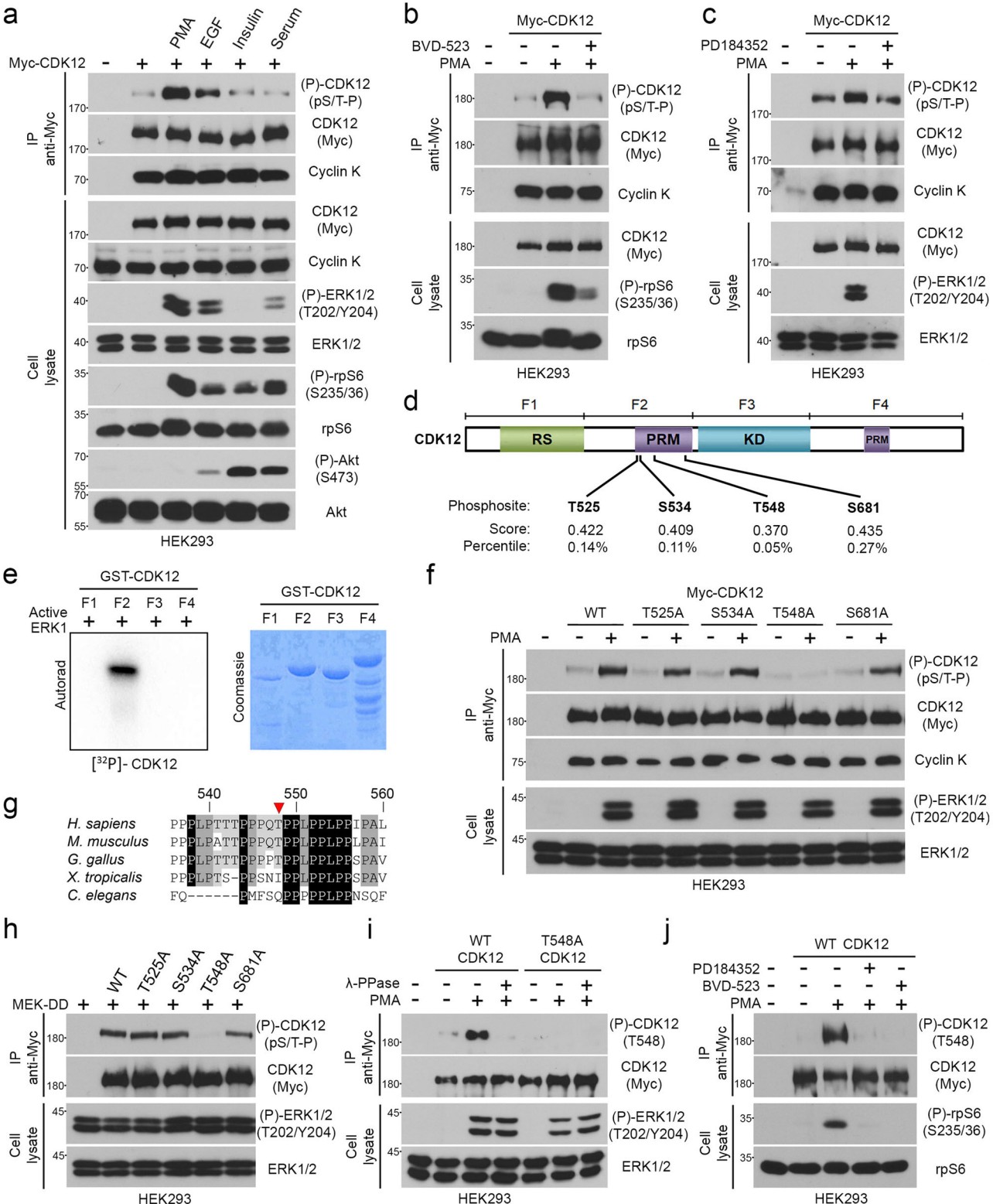

of AP-1 and NF-κB pathways in BRAF-mutated melanoma cell lines. As both pathways are involved in cancer cell growth, survival and proliferation[39,40], these results suggested that they might promote resistance to CDK12 inhibitors. Interestingly, these two pathways are enriched in very short genes (≤9.3 kb) compared to HR genes, which are themselves enriched in longer genes (≥21.8 kb) (Fig. 4m). Together, these data support the idea that CDK12 inhibition results in size-dependent gene expression that may affect drug resistance.

## JNK inhibition is synthetic lethal to CDK12 inhibition in BRAF-mutated melanoma

Transcriptome analysis revealed a significant enrichment in genes associated with the AP-1 pathway in response to CDK12 inhibition (Fig. 4k). AP-1 (Activator Protein-1) is a heterodimeric transcription factor composed of Jun and Fos isoforms, which is critically regulated by the Jun N-terminal kinases (JNKs)[41]. To determine if the AP-1 pathway contributes to CDK12 inhibitor resistance, melanoma cells (A375 and

**Fig. 2 | ERK1/2 directly phosphorylate CDK12 on Thr548. a** HEK293 cells were transfected with Myc-tagged CDK12, serum-starved overnight, and stimulated for 10–30 min with PMA, 100 ng/ml; EGF, 25 ng/ml; Insulin, 100 nM or FBS,10%. Immunoprecipitated Myc-CDK12 was assayed for phosphorylation by immuno-blotting with a phospho-motif antibody that recognizes the pS/T-P consensus motif. **b, c** HEK293 cells were transfected with Myc-tagged CDK12, serum-starved overnight, and treated with BVD-523 (2 μM) or PD184352 (10 μM), respectively, for 30 min before PMA stimulation for another 30 min. Immunoprecipitated Myc-CDK12 was assayed for phosphorylation by immunoblotting with a phospho-motif antibody that recognizes the pS/T-P consensus motif. **d** Schematic representation of human CDK12. RS arginine/serine-rich, PRM proline-rich motif, KD kinase domain. All potential ERK1/2 sites in CDK12 were predicted with high stringency using the ERK1 kinase motif of Scansite 4.0 (https://scansite4.mit.edu). Also shown is the representation of the four fragments of CDK12: F1 (aa 1–351), F2 (aa 352–711), F3 (aa 712–1101), and F4 (aa 1102–1484). **e** Recombinant active ERK1 was incubated with recombinant CDK12 (F1, F2, F3 or F4) in a kinase reaction with [γ-$^{32}$P]ATP. The resulting samples were subjected to SDS–PAGE, and the dried Coomassie-stained gel was autoradiographed. **f** Same as **b**, except that HEK293 cells were transfected with Myc-tagged CDK12 WT or unphosphorylable mutants (T525A, S534A, T548A and S681A) prior to overnight serum starvation and PMA stimulation (100 ng/ml) for 30 min. **g** Sequence alignment surrounding T548 in CDK12 from various species. **h** Same as **f**, except that cells were co-transfected with MEK-DD, an activated form of MEK1 (S218/222D), prior to being serum-starved overnight. **i** HEK293 cells were transfected with Myc-tagged CDK12 WT or an unphosphorylable mutant (T548A), serum-starved overnight, and stimulated with PMA (100 ng/ml) for 30 min. Immunoprecipitated Myc-CDK12 was then treated with 400 U λ-phosphatase for 30 min at 30 °C and CDK12 was assayed for phosphorylation by immunoblotting with a phosphospecific antibody targeted against Thr548. **j** Same as **i**, except that cells were treated with BVD-523 (2 μM) or PD184352 (10 μM) for 30 min prior to PMA stimulation (100 ng/ml) for 30 min. **a–c, e, f, g–j** Representative data of *n* = 3. Source data are provided as a Source Data file.

Colo829) were treated with increasing concentrations of THZ531 in combination with two JNK inhibitors (AS601245 and JNK-IN-8)[42]. Using the WST1 assay, we found a dramatic decrease in cell proliferation upon treatment with combinations of THZ531 and JNK inhibitors, compared to single drug treatments (Fig. 5a). Indeed, A375 cells were found to be 10- and 2.5-fold more sensitive to THZ531 when also treated with AS601245 (5 μM) and JNK-IN-8 (2 μM), respectively. Similarly, Colo829 cells were found to be 4.4- and 2-fold more sensitive to CDK12 inhibitors when also treated with AS601245 (5 μM) and JNK-IN-8 (2 μM), respectively. To determine if drug combinations displayed synergistic effects, we generated surface plots based on the Bliss independence model and found highly significant synergy (Bliss synergy score > 10) in both A375 and Colo829 cells (Fig. 5b, c). We also compared the response of melanoma cells with that of melanocytes, using both WST1 and Annexin-V binding assays to measure proliferation and apoptosis, respectively. While melanocytes were not significantly affected by the combined inhibition of CDK12 and JNK (Fig. 5d), our results show that the same combinations resulted in strong decreases in A375 (Fig. 5e) and Colo829 (Fig. 5f) cell proliferation and survival. Consistent with this, we found that combinations of CDK12 and JNK inhibitors strongly reduced long-term colony formation, compared to single drug treatments (Fig. 5g, h). Together, these data demonstrate that combinations of CDK12 and JNK inhibitors are synthetic lethal for BRAF-mutated melanoma cells.

## IKKβ inhibition is synthetic lethal to CDK12 inhibition in BRAF-mutated melanoma

Transcriptome analysis also revealed a significant enrichment in genes associated with the NF-κB pathway in response to CDK12 inhibition (Fig. 4j). As the NF-κB pathway is known to regulate cell survival in response to various cellular stresses[40], we determined if this pathway could contribute to CDK12 inhibitor resistance in melanoma cells. Using a similar approach as with JNK inhibitors (Fig. 5), melanoma cells (A375 and Colo829) were treated with increasing concentrations of THZ531 in combination with two inhibitors of IKKβ (BI605906 and MLN120B), the canonical NF-κB activator[43]. Using the WST1 assay, we observed a strong decrease in cell proliferation upon treatment with combinations of THZ531 and IKKβ inhibitors, compared to single drug treatments (Fig. 6a). Indeed, A375 cells were found to be 5.6- and 4.2-fold more sensitive to THZ531 when also treated with BI605906 (10 μM) and MLN120B (20 μM), respectively. Similarly, Colo829 cells were found to be 2.5- and 2.8-fold more sensitive to CDK12 inhibitors when also treated with BI605906 (10 μM) and MLN120B (20 μM), respectively. To determine if drug combinations displayed synergistic effects, we generated surface plots based on the Bliss independence model and found highly significant synergy (Bliss synergy score > 10) in both A375 and Colo829 cells (Fig. 6b, c). We also compared the response of melanoma cells with that of melanocytes, using both WST1

and Annexin-V binding assays to measure proliferation and apoptosis, respectively. While melanocytes were not significantly affected by the combined inhibition of CDK12 and IKKβ (Fig. 6d), our results show that the same combinations resulted in strong decreases in A375 (Fig. 6e) and Colo829 (Fig. 6f) cell proliferation and survival. Consistent with this, we found that combinations of CDK12 and IKKβ inhibitors strongly reduced long-term colony formation, compared to single drug treatments (Fig. 6g, h). Together, these data demonstrate that combinations of CDK12 and IKKβ inhibitors are synthetic lethal for BRAF-mutated melanoma cells.

To determine the potential benefit of inhibiting CDK12 and IKKβ in vivo, we first characterized the effect of THZ1[44,45], a CDK7 inhibitor that also targets CDK12 and CDK13 and that is compatible with in vivo use[46]. As demonstrated with THZ531, we found that A375 cells treated with IKKβ inhibitors are more sensitive to THZ1 (Fig. 7a), and that THZ1 and IKKβ inhibitors are strongly synergistic (Fig. 7b). To evaluate this effect in vivo, we used a xenotransplantation assay in mice. For this, A375 cells were injected subcutaneously in the flanks of NOD scid gamma (NSG) mice, which were then randomized into four groups (Fig. 7c). The different cohorts were administered twice-daily treatments of vehicle, 10 mg/kg THZ1, 50 mg/kg MLN120B, or the combination of THZ1 + MLN120B (Fig. 7c). Compared with the control group, mice treated with both inhibitors displayed rapid tumor regression (Fig. 7d). Moreover, fewer mice from the combination treatment cohort attained the defined endpoint of 200 mm³ tumor volume compared to other groups (Fig. 7e). Endpoint studies identified a reduction in cell proliferation (Ki67 positive cells) in tumors from the combination treatment cohort (Fig. 7f, g). Together, these data indicate that the combined inhibition of CDK12 and IKKβ results in significant reductions in melanoma growth in a mouse model.

## Discussion

Melanoma is the deadliest form of skin cancer due to the lack of clinical efficacy of current treatments. As the RAS/MAPK pathway is hyperactivated in melanoma and frequently involved in drug resistance, we performed a proteomics study to identify new effectors to this pathway. Using BioID, we identified 179 proteins as potential proximity interactors, from which 56 (31%) were previously found to be phosphorylated in a ERK1/2-dependent manner (Fig. 1). Amongst these, we found that CDK12 is directly phosphorylated by ERK1/2 (Fig. 2), and that the RAS/MAPK pathway positively regulates its activity (Fig. 3). We characterized the transcriptional program controlled by CDK12 in melanoma and found that CDK12 regulates gene expression based on gene length (Fig. 4). We found that CDK12 inhibition upregulates the expression of AP-1 and NF-κB gene targets (Fig. 4), and show that inhibition of these pathways sensitizes melanoma cells to CDK12 inhibition in vitro (Figs. 5 and 6) and possibly in vivo (Fig. 7). Together, these results suggest that CDK12 is hyperactivated in BRAF-

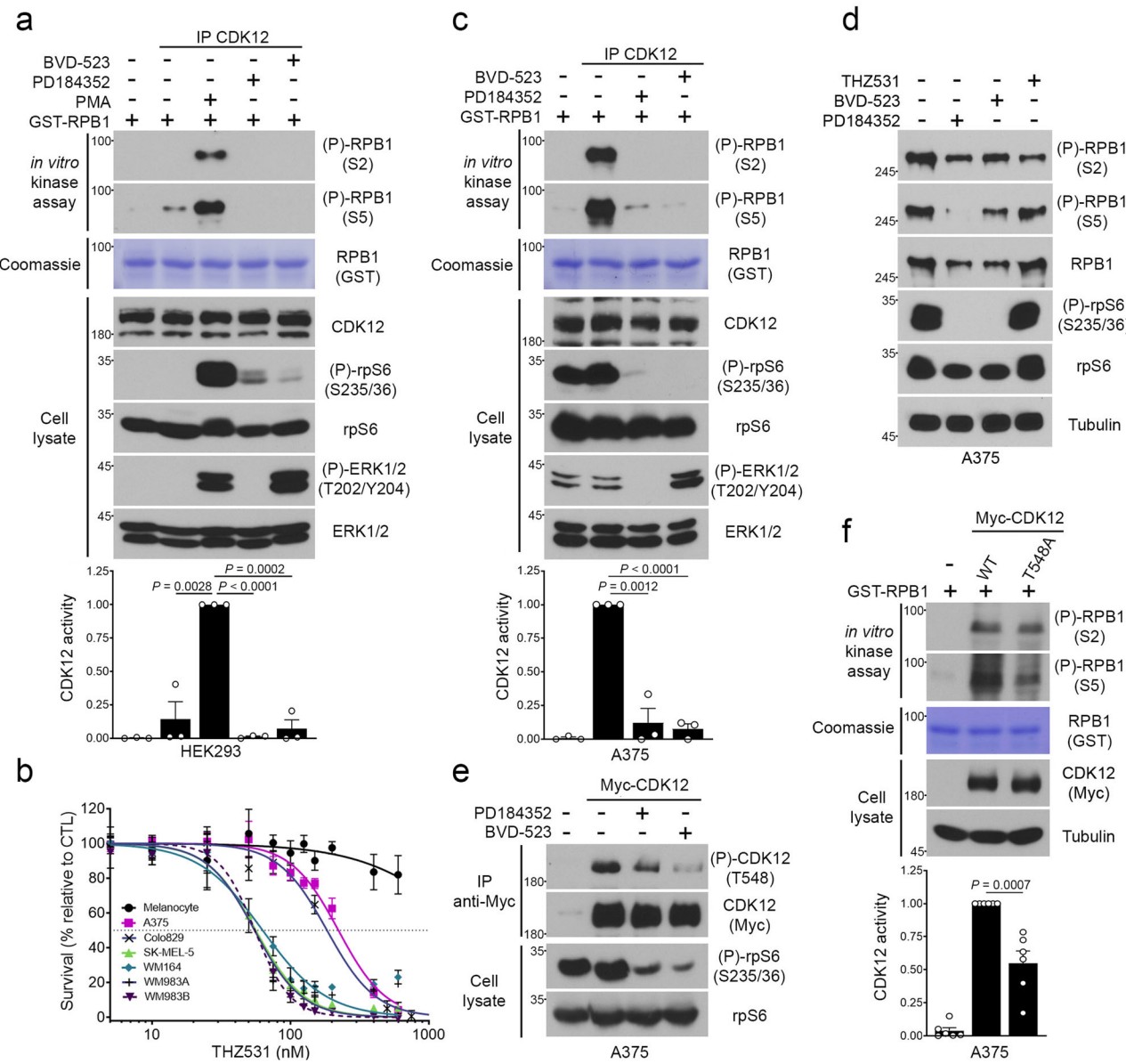

**Fig. 3 | ERK1/2 promote CDK12 activation in BRAF-mutated melanoma cells.**
**a** In vitro kinase assay based on endogenous CDK12 immunoprecipitated from HEK293 cells. Cells were serum-starved overnight, and treated with BVD-523 (2 μM) or PD184352 (10 μM) for 30 min before PMA stimulation (100 ng/ml) for another 30 min. Total cell lysates were immunoblotted with specific antibodies. For kinase reactions, recombinant GST-RPB1 was detected by Coomassie staining and samples were immunoblotted with specific RPB1 phospho-Ser2/Ser5 antibodies. CDK12 kinase activity was quantified using Ser2 phosphorylation as output. **b** Proliferation assay was performed using melanocytes and BRAF-mutated melanoma cell lines with increasing doses of THZ531 for 72 h. Respectively, the IC50 for melanocytes, >1 μM; A375, 222 nM; Colo829, 183 nM; WM164, 63 nM; WM983A, WM983B and SK-MEL-5, 55 nM. **c** Same as **a**, except that endogenous CDK12 was

immunoprecipitated from A375 cells. **d** A375 cells were serum-starved overnight, and treated with BVD-523 (2 μM), PD184352 (10 μM) or THZ531 (500 nM), for 6 h. **e** A375 cells stably expressing wild-type CDK12 (WT) were serum-starved overnight, and treated with BVD-523 (2 μM) or PD184352 (10 μM), for 1 h. Immunoprecipitated Myc-CDK12 was assayed for phosphorylation by immunoblotting with CDK12 phosphospecific antibody targeted against Thr548. **f** Same as **a**, except that CDK12 was immunoprecipitated from A375 cells stably expressing wild-type CDK12 (WT) or the CDK12 T548A mutant. Data are represented as mean ± SD of independent experiments, n = 3 (**a**, **c**), *n* = 6 (**f**) and independent replicates *n* = 6 (**b**).
**d** Representative data of independents *n* = 3. (**a**, **c**, **f**) Significance was determined using unpaired two-tailed Student's t-tests. Source data are provided as a Source Data file.

mutated melanoma and that its inhibition reveals prominent synthetic lethal targets.

Genomic alterations in the *CDK12* gene have been detected in breast[47,48], ovarian[49,50], and prostate[51,52] cancers. Downregulation or loss-of-function mutations in *CDK12* was shown to result in genomic instability and to contribute to tumorigenesis. Conversely, *CDK12* was shown to be co-amplified with *HER2* in many cases of breast cancer, and to promote tumor cell migration and invasion[53]. While both CDK12 gain- and loss-of-functions were observed in cancer, the mechanisms involved in CDK12 regulation remain unknown. Our results show that

the RAS/MAPK pathway promotes CDK12 phosphorylation and activation, and that CDK12 inhibition impairs melanoma cell survival and proliferation. As the RAS/MAPK pathway is frequently hyperactivated in cancer[54,55], our results suggest that increased CDK12 activity provides a proliferative advantage to cancer cells.

CDK12 regulates gene transcription in part by phosphorylating Pol II and thereby promoting transcription elongation[16,24]. CDK12 also regulates RNA splicing[53], and is necessary to maintain classical polyadenylation sites[20,21,56]. Alterations of CDK12 activity was shown to downregulate long genes containing multiple polyadenylation sites,

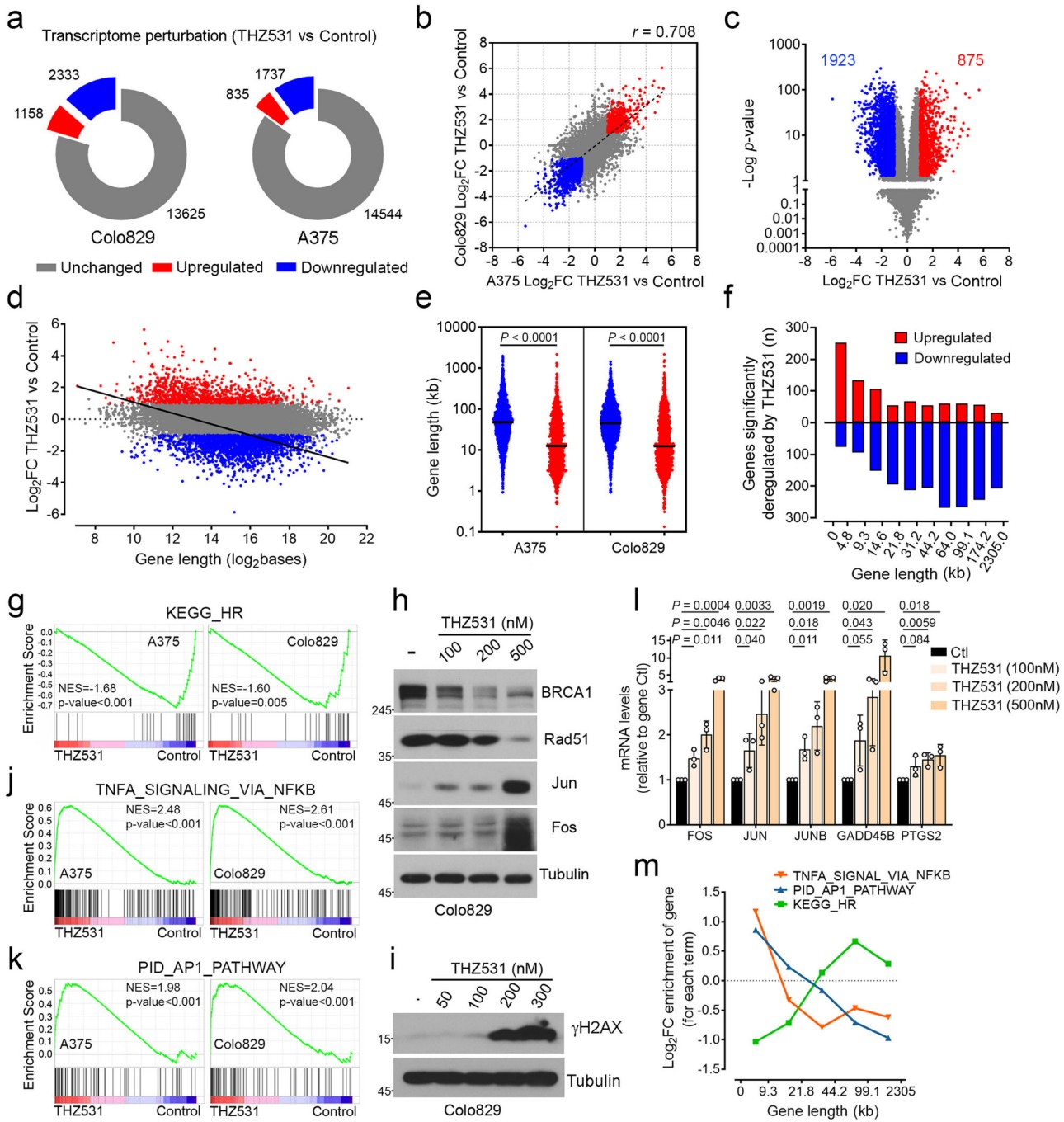

**Fig. 4 | CDK12 inhibition increases the expression of growth-promoting genes.**
**a** Global transcriptome changes associated with THZ531 (500 nM) treatment for 6 h in Colo829 (left) and A375 (right) cells. **b** Correlation between THZ531-induced gene expression changes in Colo829 and A375 cells. Displayed $r = 0.703$ value was determined using Pearson correlation. **c** Global THZ531-induced transcriptome changes (Colo829 and A375). **d** Correlation between THZ531-induced transcriptome changes in cells (A375 and Colo829) and gene length. Linear regression for significant perturbed genes is displayed (Slope −0.3297, $p$-value <0.0001). **e** Gene length of significant up- and downregulated genes by THZ531 treatment (A375, Colo829). **f** Number of genes up- or downregulated relative to gene length. The genome was ranked from smaller to longer genes and fractioned into 10 groups containing the same number of genes. **g** GSEA analysis revealed that the KEGG_HOMOLOGOUS_RECOMBINATION gene set is decreased in both THZ531-treated cells. Immunoblot of Colo829 cells treated for 12 h (**h**) or 24 h (**i**) with increasing concentrations of THZ531. **j, k** GSEA analysis revealed that the

TNFA_SIGNALING_VIA_NFKB and PID_AP_PATHWAY gene sets are positively enriched in both THZ531-treated A375 and Colo829 cells. **l** qPCR of Colo829 cells treated for 6 h with THZ531. **m** Enrichment of the number of genes according to their length for the indicated GSEA term. The genome was ranked from smaller to longer genes and fractioned into 5 groups containing the same number of genes. Data are represented as mean of $n = 3$ independent biological replicates for each cell line (**a, b, g, j, k**) and $n = 6$ (A375 and Colo829) (**c, d, f**). Length of deregulated genes from $n = 3$ independent biological replicates for each cell line (**e**). For panels (**a–f**) Log$_2$ FC (THZ531/Control) above 1 or below −1 (twofold) and $P$-values ≤ 0.05 were respectively considered as significantly upregulated (red) or downregulated (blue). (**h, i**) Representative data of independents $n = 3$. Data are represented as mean ± SD of independent experiment, $n = 3$ (**l**). Significance was determined using Wald test (**c**), or unpaired two-tailed Student's $t$-tests (**e, l**). Source data are provided as a Source Data file.

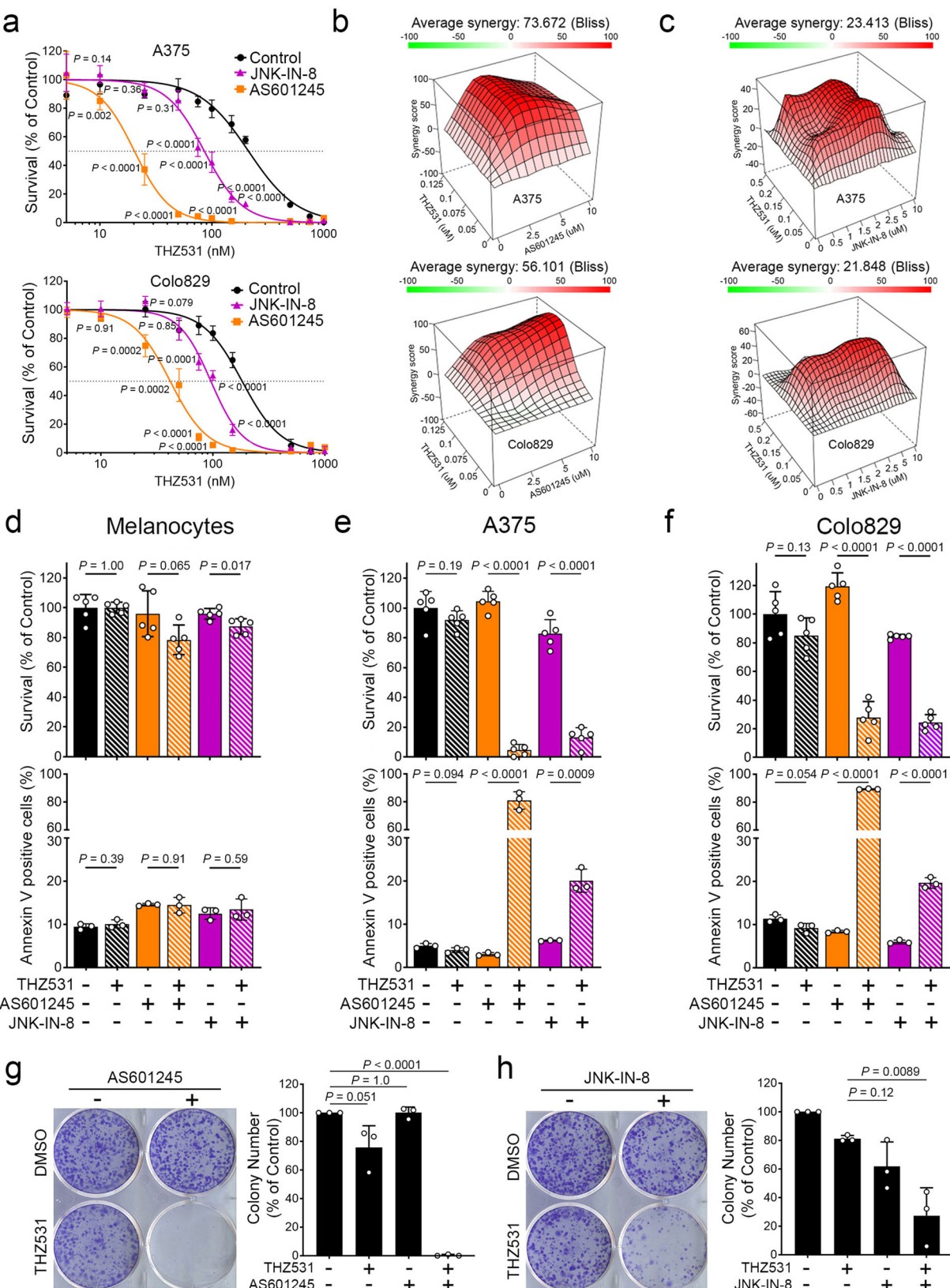

including many genes involved in the DNA damage response[21,57,58]. Accordingly, we observed a downregulation in several genes involved in homologous recombination (Fig. 4g), which was accompanied with an increase in DNA damage (Fig. 4i), as shown using γH2AX phosphorylation. Most importantly, our results indicate that CDK12 inhibition results in the overexpression of many short genes (Fig. 4f).

One potential explanation for this effect is that premature cleavage and polyadenylation of long genes contribute to Pol II recycling and accumulation at the promoter of shorter genes[59]. Among these genes, we found an increase in targets of the AP-1 and NF-κB pathways, which are involved in tumorigenesis and participate in tumor cell survival and proliferation[60–62]. These pathways were also shown to participate in

**Fig. 5 | JNK inhibition is synthetic lethal to CDK12 inhibition in BRAF-mutated melanoma. a** Proliferation assay was performed in A375 (top) and Colo829 (bottom) cells with THZ531 at indicated doses combined with AS601245 (5 μM) and JNK-in-8 (2 μM) for 72 h. Respectively IC$_{50}$ concentration for A375 with THZ531 alone 210 nM, 20 nM and 86 nM in addition with AS601245 or JNK-IN8. IC$_{50}$ for Colo829 with THZ531 alone 183 nM, 42 nM and 96 nM in addition with AS601245 and JNK-IN-8. **b, c** 3D synergy landscapes for serial dilutions of THZ531 and AS601245 (**b**) or JNK-IN-8 (**c**) combination in A375 (top) and Colo829 (bottom) cells. Representation of the Bliss synergy, Bliss scores >10 indicate drug synergy. Viability was assessed using WST1 (top) and Annexin-V (bottom) assays performed in melanocytes (**d**),

A375 (**e**), and Colo829 (**f**) cells. For WST1, THZ531 (100 nM) was combined with AS601245 (1 μM) and JNK-IN-8 (2 μM) for 72 h. For Annexin-V, THZ531 (100 nM) was combined with AS601245 (5 μM) and JNK-IN-8 (3 μM) for 48 h. Colony formation assay was performed in A375 cells with THZ531 (100 nM) combined with (**g**) AS601245 (1 μM) and (**h**) JNK-IN-8 (1 μM), for 21 days. Representation of colony formation assay (left) and quantification (right). Data are represented as mean ± SD of independent experiment, $n = 5$ (**a**), $n = 5$ for WST1 (**d–f**), $n = 3$ for Annexin-V (**d–f**) and $n = 3$ (**g, h**). Significance was determined using unpaired two-tailed Student's $t$-tests. Source data are provided as a Source Data file.

drug resistance, which is supported by our results using drug combinations in vitro and in vivo. Regarding the latter, we have used a CDK7 inhibitor (THZ1) that was also shown to target CDK12 and CDK13[46], as THZ531 is not compatible with in vivo use. Our conclusions are thus limited by the poor selectivity of THZ1 and the role of CDK7 in transcriptional regulation[44,45], which may also contribute to melanoma growth.

UV-induced DNA damage plays a key role in the initiation phase of skin cancer[63]. Consistent with this, melanomas are characterized by high levels of DNA damage compared to normal melanocytes. In this study, we found that CDK12 is active in melanoma and necessary to maintain the expression of genes involved in the DNA damage response. Consistent with this, we found that melanoma cells are more sensitive to CDK12 inhibition than melanocytes (Fig. 3b), suggesting that CDK12 activity is particularly important for the maintenance of genomic stability in melanoma. As the RAS/MAPK pathway is highly active in a large majority of melanomas, we propose that CDK12 contributes to chemoresistance by favouring the expression of genes involved in the DNA damage response. While CDK12 is a potential therapeutic target for melanoma, first-line treatments with trametinib and dabrafenib, or a combination thereof, appear much more efficient to inhibit melanoma growth in mice[64–69]. More work will be required to increase the selectivity of CDK12 inhibitors, as well as their efficacy in vivo, to evaluate their full potential as melanoma therapeutics.

## Methods

All animal procedures were performed in accordance to local animal welfare committee of the Université de Montréal (Comité de Déontologie en Expérimentation Animale, CDEA) in agreement with regulations of the Canadian Council on Animal Care (CCAC).

### Animals

NOD.Cg-Prkdc$^{scid}$ Il2rgtm1Wjl/SzJ (NSG) mice were obtained from Jackson Laboratory and bred to obtain experimental cohorts. All mice were bred and maintained under standard conditions at the Institute for Research in Immunology and Cancer. Mice were housed under specific pathogen-free conditions in filter-topped isolator cages under a 12/12 h light/dark cycle with access to food and water in all cases. All animal procedures were performed in accordance to local animal welfare committee of the Université de Montréal (Comité de Déontologie en Expérimentation Animale, CDEA) in agreement with regulations of the Canadian Council on Animal Care (CCAC).

### Cell culture

HEK293, A375, WM164 cell lines were purchased from ATCC. WM983A and WM983B cell lines were purchased from Rockland (Philadelphia, PA). Melanocytes (*CDKN2A* null) were provided by Dr. Ian Robert Watson (McGill University). All cell lines were maintained in Dulbecco's modified Eagle's medium (DMEM) (GIBCO) with 4.5 g/liter glucose supplemented with 10% fetal bovine serum (FBS), 100 IU/ml penicillin, and 100 μg/ml streptomycin, under 37 °C/5%CO$_2$ conditions. Colo829 cells were purchased from ATCC and maintained in Roswell Park Memorial Institute Medium (RPMI) with 10% FBS, 100 IU/ml penicillin

and 100 μg/ml streptomycin, under 37 °C/5%CO$_2$ conditions. Cells were regularly tested by PCR to exclude mycoplasma contamination.

### DNA constructs and recombinant proteins

The original plasmid encoding human CDK12 (pLP-3xFLAG-CDK12) was obtained from Gregg B. Morin (University of British Columbia). This DNA construct was used as PCR template for generating 6Myc-tagged CDK12 WT, T525A, S534A, T548A, and S681A in pcDNA3. CDK12 fragment 1 (F1; 1-351), F2 (352-711), F3 (712-1101) and F4 (1102-1484) were inserted in pGEX-2T for bacterial expression. The original plasmid pcDNA5-FRT/TO-FLAG-BirA$_{R118G}$ encoding Flag-tagged BirA$_{R118G}$ described previously[70] was used for generating pcDNA5-FRT/TO-FLAG-BirA$_{R118G}$-ERK1 and ERK2. The vectors encoding constitutively active MEK1 (pcDNA3-MEK-DD-Flag) was described previously[71]. pGST-CTD was provided by Dr. David L. Bentley (University of Colorado School of Medicine).

### Generation of cell line and BioID labeling

HEK293 cells transfected with bait proteins (two 150-mm plates per condition) were pelleted at low speed, washed with ice-cold PBS and lysed in 1.5 ml ice-cold RIPA buffer containing 20 mM Tris-HCl (pH 8), 137 mM NaCl, 1% NP-40, 0.1% SDS, 0.5% sodium deoxycholate, 1 mM phenylmethylsulfonyl fluoride, 1 mM dithiothreitol and a cOmplete protease inhibitor cocktail tablet (Roche), supplemented with 250U of benzonase. The lysates were sonicated using three 10 s bursts with 10 s rest in between on ice at 20% amplitude and centrifuged for 20 min. Affinity purification was performed on supernatant with 30 μl of pre-washed streptavidin-agarose beads (GE Healthcare) at 4 °C for 3 h. Then the beads were washed twice in RIPA buffer, and three times in 50 mm ammonium bicarbonate (ABC; pH 8.0). After removal of all washing buffer, beads were resuspended in 100 μl of 50 mm ABC (pH 8) with 1 μg of trypsin (Sigma) and incubated at 37 °C overnight with agitation. The next day, an additional 1 μg of trypsin was added to each sample, and the samples were incubated for 4 h at 37 °C. Beads were pelleted and rinsed two times with 100 μl of MS-grade H$_2$O. The beads supernatant and these rinses were combined, centrifuged, and the new supernatant dried in a vacuum centrifuge. Tryptic peptides were resuspended in 10 μl of 5% formic acid.

### Mass spectrometry acquisition and data analysis

Samples were reconstituted in formic acid 0.2% and loaded and separated on a homemade reversed-phase column (150 μm i.d. × 150 mm) with a 56-min gradient from 0–40% acetonitrile (0.2% FA) and a 600 nl/min flow rate on an Easy-nLC II (Thermo Fisher Scientific), connected to an Orbitrap Fusion Tribrid mass spectrometer (Thermo Fisher Scientific). Each full MS spectrum acquired with a 60,000 resolution was followed by 20 MS/MS spectra, where the 12 most abundant multiply charged ions were selected for MS/MS sequencing. Samples analyzed were converted to mzXML using ProteoWizard 3.0.4468[72] and analyzed using the iProphet pipeline[73] implemented within ProHits[74]. The database consisted of the human and adenovirus sequences in the RefSeq protein database (version 57) supplemented with "common contaminants" from the Max Planck Institute (http://maxquant.org/downloads.htm) and the Global Proteome Machine

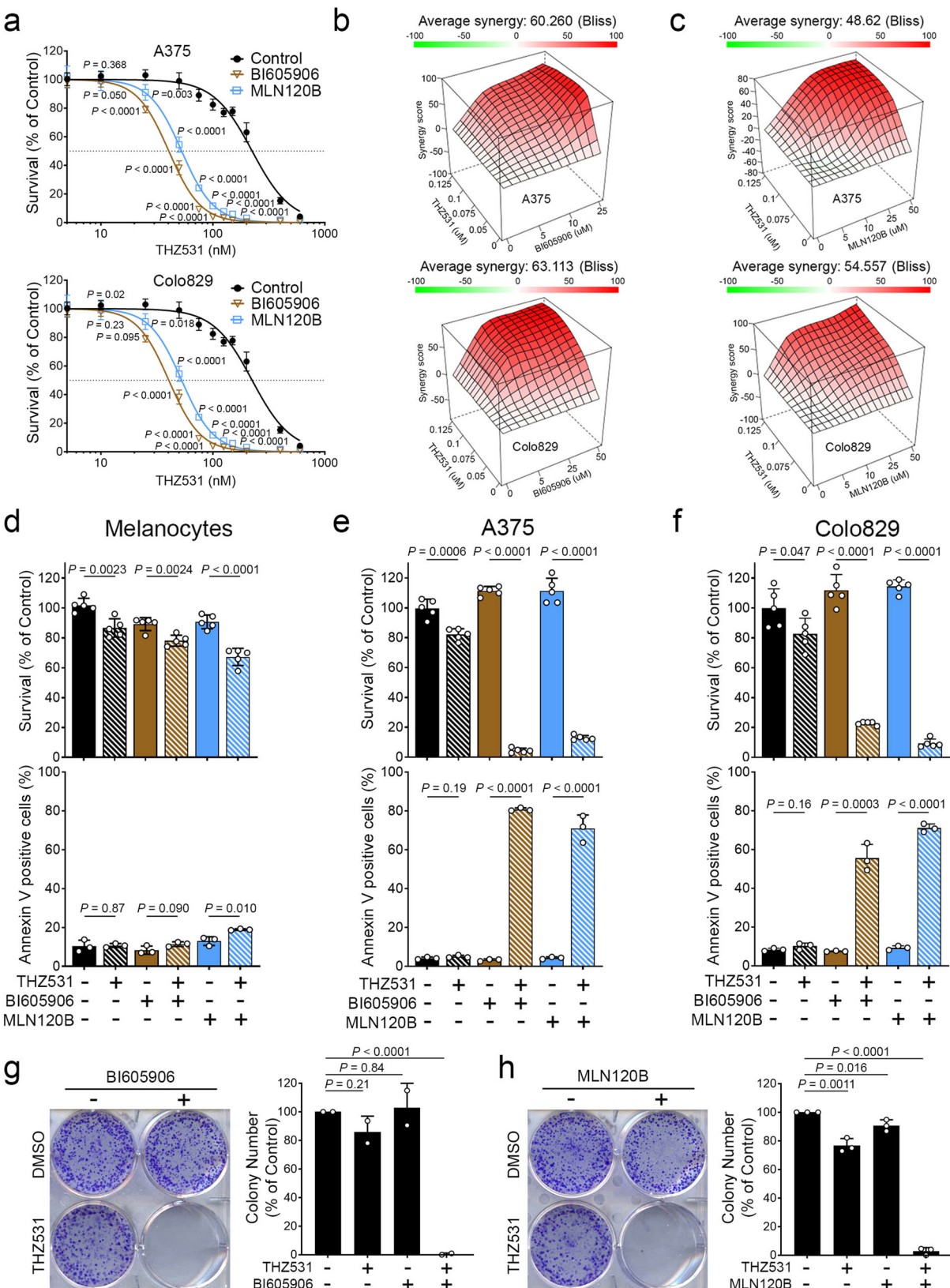

(GPM; http://www.thegpm.org/crap/index.html). The sequence database consisted of forward and reversed sequences; 72,226 entries searched. The search engines were Mascot (2.3.02; Matrix Science) and Comet (2012.01 rev.3[75], with trypsin specificity and two missed cleavage sites allowed. Methionine oxidation and asparagine/glutamine deamidation were set as variable modifications. The fragment mass

tolerance was 0.01 Da and the mass window for the precursor was ± 10 ppm. The resulting Comet and Mascot results were individually processed by PeptideProphet (42) and combined into a final iProphet output using the Trans-Proteomic Pipeline (TPP; Linux version, v0.0 Development trunk rev 0, Build 201303061711). TPP options were as follows. General options were -p0.05 -x20 -PPM -d"DECOY", iProphet

**Fig. 6 | IKKβ inhibition is synthetic lethal to CDK12 inhibition in BRAF-mutated melanoma. a** Proliferation assays were performed in A375 (top) and Colo829 (bottom) cells treated with THZ531 at indicated doses, combined with BI605906 (10 μM) and MLN120B (20 μM) for 72 h. Respectively IC$_{50}$ concentration for A375 with THZ531 alone 224 nM, 40 nM and 53 nM in addition with BI605906 or MLN120B. IC$_{50}$ for Colo829 with THZ531 alone 162 nM, 65 nM and 57 nM in addition with MLN120B. Excess over Bliss synergy plots for serial dilutions of THZ531 in combination with BI605906 (**b**) or MLN120B (**c**) in A375 (top) and Colo829 (bottom) cells. Bliss scores >10 indicate drug synergy. Viability was assessed by WST1 (top) and Annexin-V (bottom) assay was performed in

melanocytes (**d**), A375 (**e**) and Colo829 (**f**) cells. For WST1, THZ531 (100 nM) was combined with BI605906 (10 μM) and MLN120B (20 μM) for 72 h. For Annexin-V, THZ531 (100 nM) was combined with BI605906 (20 μM) and MLN120B (40 μM) for 48 h. Colony formation assays were performed in A375 cells treated with THZ531 (100 nM) combined with (**g**) BI605906 (10 μM) and (**h**) MLN120B (20 μM), for 21 days. Representation of colony formation assay (left) and quantification (right). Data are represented as mean ± SD of independent experiment, $n = 5$ (**a**), $n = 5$ for WST1 (**d–f**), $n = 3$ for Annexin-V (**d–f**) and $n = 3$ (**g**, **h**). Significance was determined using unpaired two-tailed Student's t-tests. Source data are provided as a Source Data file.

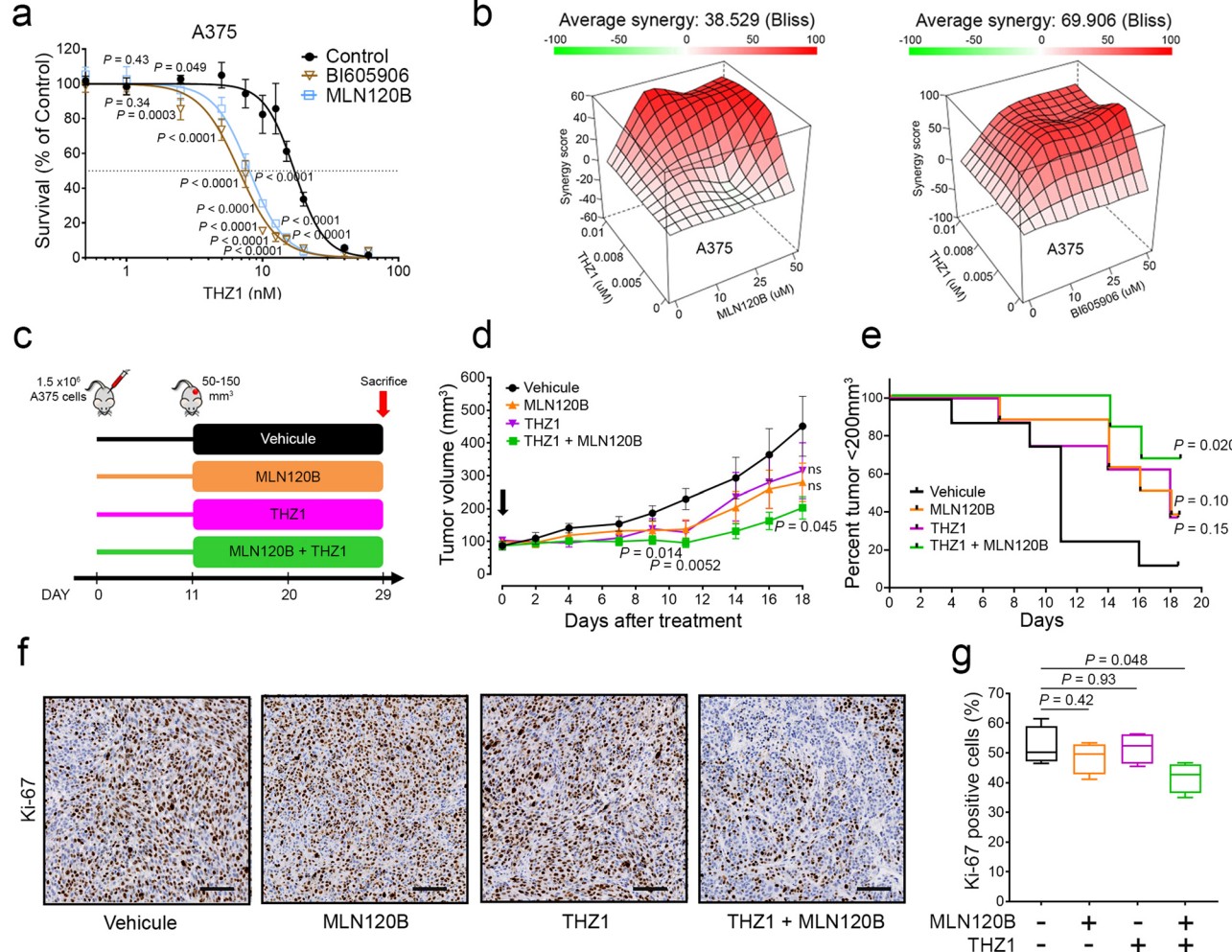

**Fig. 7 | CDK12 inhibition is potentiated by NF-κB inhibition in vivo.**
**a** Proliferation assay was performed in A375 cells treated with THZ1 at indicated doses combined with BI605906 (10 μM) and MLN120B (20 μM) for 72 h. IC$_{50}$ for THZ1 alone, in combination with BI605906 or MLN120 (17.2 nM, 6.7 nM and 6.3 nM respectively). **b** Excess over Bliss synergy plots for serial dilutions of THZ1 in combination with BI605906 or MLN120B in A375 cells. Bliss scores >10 indicate drug synergy. **c** In vivo protocol used to evaluate the role CDK12 and IKKB inhibition in BRAF-mutated melanoma cells. **d** Tumor volume measurements of an A375 xenograft mouse model treated with vehicle, 10 mg/kg THZ1 IP BID, 50 mg/MLN120B PO BID, or the combination of THZ1 + MLN120B. Treatment was stopped

on day 16. **e** For each group, percentage of tumors below 200 mm³ along time. Representative images from four different tumors (**f**) and quantifications (**g**) from immunohistochemistry profiling of mouse xenograft tissues for KI-67. **f** Scale bar: 100 μm. **g** The Tukey box and whisker plot show midline = median, box limits = Q1 (25th percentile)/Q3 (75th percentile), whiskers = 1.5 inter-quartile range (IQR). Data are represented as mean values ± SD for (**a**) and ± SEM for (**d**). $n = 6$ replicates (**a**); $n = 8$ mice for Vehicle, THZ1, MLN120B (**d**, **e**) and $n = 6$ mice for THZ1 + MLN120B (**d**, **e**); IHC from $n = 4$ mice by group (**g**). For panel (**a**, **d**, **g**), significance was determined using unpaired two-tailed Student's t-tests and Log-rank (Mantel–Cox) test for panel (**e**). Source data are provided as a Source Data file.

options were pPRIME and PeptideProphet options were pP. All proteins with a minimal iProphet protein probability of 0.05 were parsed to the relational module of ProHits. Note that for analysis with SAINTexpress, only proteins with at least two peptides identify and with iProphet protein probability ≥ 0.95 are considered. This corresponds to an estimated protein level FDR of ~0.5%. Proximity interaction

scoring was done using SAINTexpress with default parameters. Interaction data visualization was generated using Cytoscape[76]. Only proteins with Log$_2$ FC ≥ 1 (Bait/Control) and SAINT scores ≥0.80 were considered in the proximity of the bait. BirA-GFP was used as control to determine specific interactions with bait proteins (BirA-ERK1 and BirA-ERK2).

## Immunoprecipitations and immunoblotting

Cell lysates were prepared as previously described[77]. Briefly, cells were washed three times with ice-cold phosphate-buffered saline (PBS) and lysed in RIPA or BLB buffer (10 mM $K_3PO_4$, 1 mM EDTA, 5 mM EGTA, 10 mM $MgCl_2$, 50 mM β-glycerophosphate, 0.5% Nonidet P-40, 0.1% Brij 35, 0.1% deoxycholic acid), complemented with 1 mM sodium orthovanadate [$Na_3VO_4$], 1 mM phenylmethylsulfonyl fluoride, and a cOmplete protease inhibitor cocktail tablet (Roche). Lysates were centrifuged at $16,000 \times g$ for 10 min at 4 °C, supernatants were collected and heated for 10 min at 95 °C in 4× Laemmli buffer (5× buffer is 60 mM Tris-HCl [pH 6.8], 25% glycerol, 2% SDS, 14.4 mM 2-mercaptoethanol, and 0.1% bromophenol blue). Alternatively, for immunoprecipitations, supernatants were incubated with the indicated antibodies for 2 h, followed by 1 h of incubation with protein A-Sepharose CL-4B beads (GE Healthcare). Immunoprecipitates were washed three times in lysis buffer, and beads were eluted and boiled in 2× Laemmli buffer. Eluates and total cell lysates were subjected to 7.5–10% SDS–PAGE, and resolved proteins were transferred onto polyvinylidene difluoride (PVDF) membranes for immunoblotting. The data presented are representative of at least three independent experiments. Antibodies targeted against CDK12 (1/1000) and CCNK (1/1000) are from Bethyl Laboratories. Antibodies against p-ERK1/2 (E10) (T202/Y204) (1/1000), rpS6 (5G10) (1/1000), p-rpS6 (S235/236) (1/1000), AKT, p-AKT (S473) (D9E) (1/1000), RPB1 (4H8) (1/1000), p-RPB1 (S2) (E1Z3G) (1/1000), p-RPB1 (S5) (D9N5I) (1/1000), p-H2AX (S139) (20E3) (1/750), GST (1/500), Jun (60A8) (1/1000), BRCA1 (1/750), Rad51 (D4B10) (1/1000); p-RSK1 (1/1000) and p-MAPK Substrates Motif [PXpTP] (1/1000) were purchased from Cell Signaling Technologies. Antibodies against Tubulin (T5618) (1/4000), Actin (A5441) (1/2000), Myc (9E10) (1/2000), Flag (2EL-1B11) (1/1000) from Millipore Sigma; c-Fos (H-125) (1/500) and RSK1 (C-21) (1/500) from Santa Cruz Biotechnology; HA (12CA5) (1/1000) from ThermoFisher; and p-CDK12 (T548) (1/750) was generated with AMSBIO (Cambridge, MA). While we found that the p-CDK12 (T548) antibody specifically recognizes phosphorylated CDK12, our results show that this antibody could only be used against immunoprecipitated, overexpressed, protein. All secondary horseradish peroxidase (HRP)-conjugated antibodies (Anti-Rabbit and Anti-Mouse) (1/10,000) used for immunoblotting were purchased from Millipore Sigma.

## Survival assays

Cells were grown in medium supplemented with 10% FBS and relative number of viable cells was measured using WST1 reagent from Millipore Sigma. WST1 was added 2 h prior to absorbance measurement at 450 nm using a Tecan Infinite M200 PRO microplate reader and TECAN i-control software. Cells were plated and after adhesion were treated with THZ531 (5–1000 nM or 100 nM) or THZ1 (0.5–60 nM) in addition to BI601245, MLN120B, AS601245 or JNK-IN-8 at the indicated concentration. Cells were grown for 72 h before addition of WST1. The results displayed represent the mean of $n = 5$ or $6 \pm$ standard deviation (SD). Bliss scores were calculated using SynergyFinder 2.0[78].

## Annexin-V apoptosis assay

Cells were seeded in six-well dishes and grown in medium supplemented with 10% FBS. After 24 h, cells were treated with THZ531, BI601245, MLN120B, AS601245, or JNK-IN-8 at the indicated concentration for 48 h. Cells were collected and washed twice in cold PBS, stained with PE-Annexin-V (BD Biosciences), according to the manufacturers' instructions before flow cytometry analysis. Samples were acquired using BD FACS Canto II instrument and BD FACS Diva v8.0.2 software. Gating strategy is provided in Supplementary Fig. 4. Data were represented as Mean Fluorescence Intensity (MFI) or percentage (%) of Annexin-V-positive cells.

## RNA-sequencing and transcriptome analysis

The transcriptomes of A375 and Colo829 cells treated with DMSO or THZ531 (500 nM, 6 h) were analyzed by RNA-seq (corresponding to Supplementary Data 2). Total RNA was isolated from cells using RNeasy mini kit (Qiagen), according to the manufacturer's instructions. RNA was quantified using Qubit (Thermo Scientific) and quality was assessed with the 2100 Bioanalyzer (Agilent Technologies). Transcriptome libraries were generated using the Kapa RNA HyperPrep (Roche) with a poly-A selection (Thermo Scientific). Sequencing was performed on the Illumina NextSeq500, obtaining around 30 M 75 bp single-end reads per sample. Library preparation and sequencing was done at the Institute for Research in Immunology and Cancer (IRIC) Genomics Platform. For mapping, sequences were trimmed for sequencing adapters and low quality 3' bases using Trimmomatic version 0.35 and aligned to the reference human genome version GRCh38 using STAR version 2.7.1a. Expression levels of mRNA were displayed as reads per kilobase per million (RPKM). Data are representative of three independent biological experiments. RPKM values of each biological replicate were averaged for Control and THZ531. Only transcript with $Log_2$ FC $\geq 2$ (THZ531/Control) or $Log_2$ FC $\leq 2$ (THZ531/Control) and ($P < 0.05$), were respectively considered upregulated or downregulated by CDK12 inhibition (THZ531 treatment).

## Viral Infections and transfection

For stable cell line generation, lentiviruses were produced in the HEK293T cell line using the pLenti-CMV-GFP-puro vector to overexpress ectopic GFP, CDK12 WT and T548A mutant. For shRNA-mediated knockdown of CDK12, lentiviruses were produced using vectors from the Mission TRC shRNA library (Sigma-Aldrich) targeting CDK12 (#1 TRCN0000001795, #2 TRCN0000001797, #3 TRCN0000001798). Two days after infection, A375 were selected with 2 µg/ml puromycin. HEK293 and HEK293T cells were transfected by calcium phosphate precipitation. Briefly, plasmid and $CaCl_2$ 2 M were diluted v/v in 2× HBS (274 mM NaCl, 1.5 mM $Na_2HPO_4$, 55 mM HEPES, pH 7) before adding slowly to cells.

## Xenograft assays in NSG mice

Male NSG (18-week old) were enrolled in the experimental cohorts. A375 cells underwent screening to confirm that the cell line was free of rodent infectious agents (Mouse essential clear panel from Charles River). Cells were administered ($1.5 \times 10^6$ cells) subcutaneously into the right flank in a volume of 100 µl. A375 cells were suspended in 50 µl of phosphate buffer solution and mixed in a 1:1 ratio with Matrigel prior to injections. When the tumor size reached ~120 mm³, mice were randomized into four groups ($n = 8$ per group): Vehicle, THZ1 (10% DMSO, D5W), MLN120B (Methylcellulose 0.5%) and THZ1 + MLN120B combination. THZ1 was administered IP b.i.d. at 10 mg/kg, and MLN120B was administered PO b.i.d. at 50 mg/kg. Mice were treated 5 days per week during 18 days, and tumors were measured three times per week. After treatment, mice were sacrificed using a $CO_2$ chamber and tumor tissues were excised and fixed immediately for IHC analysis. None of the tumors reached a volume that exceeded the humane endpoint value of 1500 mm³.

## Immunohistochemistry

IHC assays were performed on formalin-fixed paraffin-embedded (FFPE) tissue samples. These assays were carried out according to the manufacturer recommendations on an automated IHC stainer Bond RX (Leica Biosystems, Buffalo Grove, IL, USA). IHC analysis was performed against Ki-67 [Biocare Medical; CRM325B, rabbit monoclonal, dilution 1/150], using the BOND Epitope Retrieval Solution 1 (ER1; citrate based, pH 6) for 20 min for heat-induced Epitope Retrieval. Sections were incubated with 150 µl of antibody at RT for 30/15 min (primary/secondary antibody, respectively). Detection of specific signal was acquired by using Bond Polymer Refine Detection System

containing the DAB chromogen (#DS9800, Leica Biosystems). Stained slides were scanned using the NanoZoomer Digital Pathology (NDP) 2.0-HT digital slide scanner (Hamamatsu, Japan). IHC quantifications were performed in 4 tumors/group.

## CDK12 kinase assay

Cells were serum-starved overnight, and treated with BVD-523 (2 μM) or PD184352 (10 μM) for 30 min. before to PMA stimulation (100 ng/ml) for another 30 min. Cells were lysed in RIPA buffer and CDK12 immuno-precipitates were washed thrice in lysis buffer followed by three washes in kinase buffer (25 mM Tris-HCl [pH 7.4], 10 mM $MgCl_2$, 34 mM KCl, and 5 mm β-glycerophosphate). Then, recombinant RPB1 purified from BL21(DE3)pLysS cells was used as substrates with immunoprecipitated CDK12 as kinase. Assays were performed for 60 min at 30 degrees in kinase buffer (with 200 μM ATP), and then stopped by adding 2× reducing sample buffer. Eluates and total cell lysates were subjected to 7.5% SDS–PAGE, and resolved proteins were transferred onto poly-vinylidene difluoride (PVDF) membranes for immunoblotting.

## Colony formation assays

A375 were seeded into six-well plates ($1 \times 10^4$ cells per well) and grown for 10 days (fed every 3 days) in medium supplemented with 10% FBS in the presence of drugs as indicated. Cells were fixed with methanol and stained with 0.5% crystal violet (in 25% methanol). Colonies were counted using Image J software.

## RNA extraction and real-time quantitative-PCR (qPCR)

Total RNA was extracted using RNeasy mini Kit (Qiagen) and reverse-transcribed using the cDNA Reverse Transcription Kit (Applied Bio-systems), as described by the manufacturer. For each qPCR assay, a standard curve was performed to ensure that the efficacy of the assay (between 90 and 110%). The Viia7 qPCR instrument (Life Technologies) was used to detect amplification level and was programmed with an initial step of 20 s at 95 °C, followed by 40 cycles of 1 s at 95 °C and 20 s at 60 °C. Relative expression ($RQ = 2^{-\Delta\Delta CT}$) was calculated using the Expression Suite software (Life Technologies), and normalization was done using both *ACTB* and *GAPDH*. Primer sequences for qPCR analysis are provided in Supplementary Table S1.

## Quantification and statistical analysis

For BioID, a SAINT score was calculated to determine the ERK1/2 proximity interactome. SAINTScore ≥ 0.8 and $Log_2$ FC ≥ 1 were considered as cut-off values. *r* correlations were determined by Pearson correlation. For all relevant panels unless specified, center values and error bars represent means ± SD and *p*-values were determined by unpaired two-tailed Student's *t*-tests used for comparisons between two groups with at least $n = 3$. Figures 4e and 7d; *P*-values were determined by one-way ANOVA tests. Figure 7e; *P*-values were determined by Log-rank (Mantel–Cox) test. All experiments were performed a minimum of three times. Statistical details for each experiment are located in the results, figures, and figure legends.

## Data availability

The raw datasets of BioID (Supplementary Data 1) and transcriptomics (Supplementary Data 2) generated for this study are available as supplementary information in excel sheet formats. BioID data (corresponding to Supplementary Data 1) have been deposited in ProteomeXchange through partner MassIVE as a complete submission and assigned MSV000087994 (https://massive.ucsd.edu/ProteoSAFe/dataset.jsp?task=87751974c6e943bf8c8ecd85bfa27259) and PXD027983. The data can be downloaded from ftp://massive.ucsd.edu/MSV000087994/. Gene signatures used for GSEA analyses presented in Fig. 4g, j, k are available at https://www.gsea-msigdb.org/gsea/msigdb/cards/KEGG_HOMOLOGOUS_RECOMBINATION, https://www.gsea-msigdb.org/gsea/msigdb/cards/PID_AP1_PATHWAY and

https://www.gsea-msigdb.org/gsea/msigdb/cards/HALLMARK_TNFA_SIGNALING_VIA_NFKB.html, respectively. RNA-seq raw data generated for this study have been deposited in the GEO under accession number GSE184734. The remaining data are available within the Article, Supplementary Information or Source Data file. Source data are provided with this paper.

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

## Acknowledgements

We thank all members of the laboratory for their insightful discussions and comments. We thank Dr. David L. Bentley (University of Colorado School of Medicine) for providing pGST-CTD, Dr. Gregg B. Morin (University of British Columbia) for providing pLP-3xFLAG-CDK12, and Dr. Ian R. Watson (McGill University) for providing control melanocytes. We are grateful to Dr. Anne-Claude Gingras (University of Toronto) for the use of ProHits. We also acknowledge technical assistance from Mélania Gombos and the personnel at the IRIC animal facility for mouse xenograft experiments. We also hugely benefited from the expertise of personnel at IRIC's Genomics and Histology Core Facilities for RNA and immunohistochemistry experiments, respectively. This work was supported by grants from the Cancer Research Society (to P.P.R.), the Canadian Institutes for Health Research (to P.P.R. and S.M.), and from the Natural Sciences and Engineering Research Council of Canada (to P.P.R.).

## Author contributions

Conceptualization: T.H., S.M, and P.P.R.; Methodology: T.H., G.L., M.K.S., and P.P.R.; Investigation: T.H., G.L., S.N., W.C., E.V., C.G. M.B., B.G., S.L., M.K.S and P.P.R.; Writing: T.H. and P.P.R.; Funding acquisition: S.M., S.A., and P.P.R.; Supervision: S.A., S.M., and P.P.R.

## Competing interests

The authors declare no competing interests.
