## [Peer Review File · Nature Communications]

Reviewers' Comments:

Reviewer #1:

Remarks to the Author:

The paper by Houtes et al finds CDK12 to be a downstream target of RAS/MAPK signalling pathway, that is largely responsible for drug resistance during melanoma treatment. The authors identify threonine 548 (Thr548) in CDK12 proline-rich region as a substrate of ERK1/2 kinases and propose this signalling to be responsible for CDK12 hyperactivation in BRAF-mutated melanoma cells. Using covalent CDK12/CDK13 inhibitor THZ531 they show that treatment of BRAF-mutated melanoma cells results in downregulation of long genes (particularly DNA-repair ones) and upregulation of short genes including many components of pro-growth AP-1 and NF-KB pathways. Finally, they show that inhibition of these two pathways synergizes with CDK12 inhibition and results in growth suppression of melanoma cells. The paper shows important findings, particularly potential link between RAS/MAPK pathway and CDK12 and synthetic lethality between CDK12 and AP-1/NF-KB pathways will be of a broad scientific interest. I would support publication in Nature Communications after the following concerns are addressed.

Major comments

1) Fig. 3: It is not clear why P-Ser7 is phosphorylated in the GST-CTD purified from bacteria (Fig. 3 b, e, g, Supp. Fig. 1a). Proteins expressed in bacteria are not phosphorylated. This would also explain why the P-Ser7 band is not sensitive to the THZ531 treatment (Supp. Fig. 1). Is the P-Ser7 band phosphatase-sensitive? Thus, the results of the experiment can not be used to conclude that CDK12 phosphorylates Ser2 and Ser5, but not Ser7.

There is a relatively strong evidence that CDK12 is a promiscuous CTD kinase in human cells. This conclusion is supported by various in vitro kinase assays (1,2) and by selective inhibition of CDK12 in cell lines carrying analog-sensitive CDK12 alleles (3-6). These papers show small and distinct effects of CDK12 on the CTD phosphorylation and the results should be considered when roles of CDK12 in the CTD phosphorylation and transcription are introduced. Another issue in the manuscript are possible off-target effects of higher concentration of THZ531 (>100 nM) (7). Lower concentrations of THZ531 would be better to secure selective inhibition of CDK12 (see also comment in points 7 and 8). All these issues are reviewed for example here (8) and should be considered in experimental design and data interpretation.

2) Fig. 3b, e: If change in in vitro activity is a result of (or correlates with) phosphorylation of Thr548 this needs to be presented on the western blots. This is an important control to support conclusion of the paper.

3) Fig. 3b, lane 2 and Supp. Fig. 1a, lane 3: Discrepancy between the figures - "not activated" CDK12 shows strong phosphorylation of Ser2 in Supp. Fig.1a, but no phosphorylation in Fig.3b.

4) Fig.3c: Can loading controls other than RBP1 and ERK1/2 be shown? It is hard to compare between cell lines because the RBP1 levels differ significantly. The P-Ser5 and P-Ser7 levels need to be also shown (see point 1). In addition, what are the levels of total CDK12 protein and P-Thr548 in these cell lines? If P-Ser2 increase (or possibly P-CTD increase?) is due to the hyperactivated CDK12 (line 174 and title of the paper) this should be documented by a corresponding increase in P-Thr548.

5) Does ERK1/2 activation/inhibition affect the global state of the CTD phosphorylation (P-Ser2,5,7)? This should be tested in total cell lysates of selected melanoma cell lines.

6) I would suggest to use expression of well-characterized CDK12-dependent DNA-repair/replication genes as a readout of CDK12 activity after ERK1/2 inhibition/activation to make a stronger case for the role of ERK1/2 in CDK12 activation. This should give much clearer picture than solely relying on in vitro kinase assays with the CTD (with all the caveats described above).

7) Fig. 4: RNA-seq was performed with relatively high concentrations of THZ531 (500 nM). To exclude a possibility of upregulation of genes in AP-1 and NF-KB pathways due to off-target

effects, RT-qPCR with lower concentrations of THZ531 and/or another CDK12 inhibitor SR-48359 should be performed. Increase in protein levels of the components of AP-1 and NF-KB pathways (Fos, Jun, Gadd45a or others) should be documented by western blotting.

8) Fig. 7: THZ1 is primarily a CDK7 inhibitor with some degree of affinity towards CDK12 and CDK13 (10). At minimum, this should be clearly stated in the text. CDK7 activates many other kinases, particularly CDK9, CDK1, CDK2, CDK4, CDK6 (11,12) and this can also significantly affect the results. Did authors consider using other CDK12 inhibitor such SR-48359 if THZ531 can't be used in in vivo studies? This compound was used in vivo models (9).

9) Discussion, lanes 292, 320: Additional experiments (see above) are required to confirm the hyperactivated state of CDK12 in melanoma

Minor comments

1) Fig. 1b and line 99: the GFP biotinylation signal seems to be stronger in comparison to ERK1/2.

2) Fig. 1: It is not clear if HEK293 empty vector (EV) or GFP-BirA cells (or both) were used as a control for the MS data analysis.

3) Line 121: It would be informative to show figure documenting conservation of Thr548 in CDK12 across species.

4) Fig. 2: EGF abbreviations should be spelled out, use of serum starvation experiment should be briefly explained as well as meaning of phospho-markers (P-ERK1/2, (P)-rpS6, P-Akt-S473...). Anti-pS/T-P antibody is not described in the materials and methods. They should also explain why they use overexpressed CDK12 and not the endogenous one to show phosphorylation of Thr548.

5) Fig. 4h: concentrations of THZ531 should be specified.

6) Fig. 7b: 2nd graph – replace MLN120B with BI601245.

7) Methods:

Lane 349: Correct HEK293 BirA-ERK1/ERK2; A375 cells stably expressing Myc-CDK12 (WT/Thr548Ala) should be also described.

Lane 368: Correct to "Stable Inducible Cell Line"

Lane 440: Delete section "Immunofluorescence Microscopy" (not used in the paper).

Lane 507: Delete Immunohistochemistry– Caspase-3 antibody (not used in the paper).

Lane 527 : Correct "with immunoprecipitated CDK12 as kinase"

References:

1. Bosken, C.A. et al. The structure and substrate specificity of human Cdk12/Cyclin K. *Nat Commun* 5, 3505 (2014).
2. Bartkowiak, B. & Greenleaf, A.L. Expression, purification, and identification of associated proteins of the full-length hCDK12/CyclinK complex. *J Biol Chem* 290, 1786-95 (2015).
3. Chirackal Manavalan, A.P. et al. CDK12 controls G1/S progression by regulating RNAPII processivity at core DNA replication genes. *EMBO Rep* 20, e47592 (2019).
4. Tellier, M. et al. CDK12 globally stimulates RNA polymerase II transcription elongation and carboxyl-terminal domain phosphorylation. *Nucleic Acids Res* 48, 7712-7727 (2020).
5. Bartkowiak, B., Yan, C. & Greenleaf, A.L. Engineering an analog-sensitive CDK12 cell line using CRISPR/Cas. *Biochim Biophys Acta* 1849, 1179-87 (2015).
6. Fan, Z. et al. CDK13 cooperates with CDK12 to control global RNA polymerase II processivity. *Sci Adv* 6(2020).
7. Zhang, T. et al. Covalent targeting of remote cysteine residues to develop CDK12 and CDK13 inhibitors. *Nat Chem Biol* 12, 876-84 (2016).
8. Pilarova, K., Herudek, J. & Blazek, D. CDK12: cellular functions and therapeutic potential of versatile player in cancer. *NAR Cancer* 2, zcaa003 (2020).
9. Quereda, V. et al. Therapeutic Targeting of CDK12/CDK13 in Triple-Negative Breast Cancer.

Cancer Cell (2019).

10. Kwiatkowski, N. et al. Targeting transcription regulation in cancer with a covalent CDK7 inhibitor. *Nature* 511, 616-20 (2014).

11. Larochelle, S. et al. Cyclin-dependent kinase control of the initiation-to-elongation switch of RNA polymerase II. *Nat Struct Mol Biol* 19, 1108-15 (2012).

12. Schachter, M.M. et al. A Cdk7-Cdk4 T-loop phosphorylation cascade promotes G1 progression. *Mol Cell* 50, 250-60 (2013).

Reviewer #2:

Remarks to the Author:

Houles et al performed proximity labeling using ectopic BirA-ERK1 and 2, cross referenced their results with two other published datasets and came up with a list of 23 proteins of interest, of which they chose to focus on CDK12. The authors nicely demonstrate by multiple means that ERK1/2 phosphorylate CDK12 on Thr548, including going so far as to generate their own phospho-specific antibody for this site and validating this phosphorylation by epistatic analysis. Nice work! The authors then provide evidence that PMA-induced phosphorylation of CDK12 promotes its kinase activity based on an IP-kinase assay using the known substrate, the CTD domain of a pol II subunit. Again, these are nice experiments. To tie this into a function of CDK12, the authors performed RNAseq on two cell lines with high MAPK activation, plus and minus a CDK12 inhibitor. This led to a series of drug synergy experiments with JNK and IKKbeta inhibitors, which is fine and adds some clinical relevance to the study, but in terms of mechanism is underdeveloped and somewhat over-interpreted. The authors then take it home with an in vivo study, showing some decrease in tumor growth when a CDK12 inhibitor is combined with an IKKbeta inhibitor. All in all, this is a great study, especially the first half. I also appreciate that the authors add clinical relevance to their work by performing RNAseq analysis to find a possible downstream druggable target to combine with a CDK12 inhibitor. Concerns are all addressable. Recommend the authors have the opportunity to revise their manuscript for resubmission.

Major concerns

-Key to the success of BirA proximity labeling is validation that the fusion protein actually functions correctly. Please confirm that BirA-ERK1 reverts a null phenotype (e.g. restores proliferation, restores transcriptional signatures as assessed by whole genome RNAseq analysis, restores the phosphorylation of ERK substrates by whole genome phospho-proteomics, or whatever comprehensive phenotype the authors choose) of ERK1 sgRNA-treated cells validated to lack ERK protein expression, please include the negative control (BirA alone) and positive control (unfused ERK1) expression constructs. Repeat the same experiment with BirA-ERK2. Please note that this is a critical requirement for publication, and thus the authors are strongly encouraged to perform these experiments.

-The RNAseq analysis is somewhat over-interpreted, as this is really analysis of a CDK12 inhibitor (plus no data was provided that the inhibitor works in the cell assayed) and not CDK12 activity per se. What I mean by this is would you expect the same profile in cells with low CDK12 phosphorylation? Is this a dataset of CDK12 function in general, rather than an activated response? Either rephrase the interpretation of this experiment or cross reference it with treatment of melanoma line without high CDK12 phosphorylation with the CDK12 inhibitor, or if none exist, then some other cell line with low CDK12 phosphorylation. Again, the point here is whether the authors are looking at the effect of high CDK12 activity due to robust MAPK activation, or the effect of CDK12 function in general.

-Please consider softening the conclusions on the synergy experiments, as classic epistatic analysis was not performed to validate the mechanism of this synergy. While I have no problem with the idea of the RNAseq analysis supporting drug synergy experiments, I would caution the authors on over-interpreting these in terms of a direct connection to CDK12 function, as one could imagine grossly disrupting the mRNA pool with a CDK12 inhibitor could set up all kinds of potential synthetic lethality, and JNK and IKKbeta are pretty broad in their effects.

-The authors' line of investigation in figure 7 suggests that CDK12 may offer a new druggable kinase to treat melanoma, indeed this was the impetus for the study as described in the introduction, yet the effect on xenograft growth appears rather modest. As a key control to put this work into context with current therapies, please repeat Fig 7d,e, but instead test with

vemurafenib (or dabrafenib) AND trametinib. This is a really important control, as it will help inform on the possible utility of this new drug synergy in the clinic.

-Please test whether the Phospho-Thr548 antibody detects high CDK12 phosphorylation on clinical melanoma samples, as this would go a long way to supporting the contention that CDK12 is upregulated during melanomagenesis. A small panel of tumors is fine, as the study is already quite comprehensive.

Minor concerns

-Please move Fig 1a to the supplement, as at this point proximity labeling is a well-known assay

-Please clarify that the BirA-ERK fusion proteins are ectopically expressed, and please compare the level of these proteins to the endogenous ones. There is no problem with using ectopic BirA-ERK, others have done this with great success with other BirA fusion proteins, but it is helpful to know the expression level, especially in terms of comparing the interactome to other datasets.

-please either normalize the BirA-ERK dataset against one generated with BirA alone (in the presence of biotin), or note in the results that the dataset was not normalized to BirA alone, as this is a common control for these types of experiments.

-the authors claim that serum does not lead to CDK12 phosphorylation because it activates the PI3k/AKT pathway, yet ERK is clearly phosphorylated, please consider revising the text to better reflect the results, for example, this result suggests that potent MAPK activation leads to CDK12 phosphorylation.

-please include the full-length gels in the supplement, this is particularly important to gauge how specific the antibodies are, for example, for the new Phospho-Thr548 antibody.

-please explicitly state in the figure legends the number of replicates and type (technical or biological) for each experiment, and when multiple experiments have been performed, please consider reporting the averages in a graph in the supplemental data. I leave this to the authors to decide which experiments to quantitate, but minimally all replicates need to be included in the supplemental data as full-length gels and so forth. This simply provides one metric to help gauge the degree of rigor for each experiment.

-Is there an active ERK mutant the authors could use to validate phosphorylation on Thr548 in cells, analogous to the experiment in Fig 2g? Such an experiment would further strengthen their claims.

-Fig 3a is not necessary, these are pretty standard approaches, so please move to the supplement.

-T>D mutants can sometimes act as phosphomimics, please generate a Thr548>Asp CDK12 mutant and determine whether it has increased kinase activity in vitro and in cells. If positive, this would clearly be nice supporting data, if negative, the authors have ruled out an obvious independent test of their hypothesis.

-Fig 7c really is not required, please move to supplement

-please speculate on the relatively poor overlap between proximity labeling and the other approaches described in Fig 1f in the discussion

-various typos in the manuscript, especially the materials and methods that need to be corrected.

Reviewer #3:

Remarks to the Author:

In their paper Houles and co-workers explore the role of CDK12 in BRAF mutant melanoma. This is generally a well written paper but I do have several suggestions/recommendations below:

1. What is a "prime" synthetic lethal target? The word "prime" tells me nothing about the role of CDK12 in the fitness of BRAF-mutant melanoma cells. I'd suggest re-working the title.

2. I think Figure 1A could be improved. Even with my big screen I can't see all of the writing so I can't imagine it will be visible when printed. Maybe this part of the figure could be enlarged and moved to the supplementary data.

3. Error in figure legend 1c should read 0.918 not 0918.

4. In figure 1e – I can see that ERK1/2 interactors are grouped. Does this grouping have any statistical significance? The suggestion is that there is an enrichment for factors involved in (for example) transcription, cell trafficking etc but that's not what this figure tells me. Could this be clarified please?

5. I like the way the authors have used multiple approaches to end up with 31 interactors. This is

a good strong approach. Were there any established interactors that were missed? Can they give the readers a sense for the false negative rate?

6. Line 118 – I would say “23 novel potential ERK1/2 substrates....”

7. As a melanoma biologist/geneticist I got a bit lost when THZ531 was introduced. Could you please explain what it is and why you reasoned that it would abrogate CDK12-mediated phosphorylation? I found the introduction of this compound somewhat abrupt.

8. As an aside the other place where the MAPK pathway is activated via BRAF mutation is in paediatric low grade glioma.

9. I wonder if the authors could explain why CDK12 is not an essential gene in the all of the BRAF mutant melanoma cell lines CRISPR screened by DepMAP:

<https://depmap.org/portal/gene/CDK12?tab=overview> or alternatively use these data to further explore the role of CDK12 in this disease?

10. What happens to the expression of long and short genes when cells are sick? I guess I wonder if the correlations reported (e.g. gene length relative to CDK12 activity) are really a read out of cell fitness rather than being a phenotype (selectively) associated with perturbation of CDK12.

11. Figure 5 b,c, Fig 6bc and fig7b are (I’m sorry) a bit meaningless. I really don’t understand the message of these figures. They all look the same to me.

12. Figure 7f needs a scale bar.

Overall this paper includes an enormous amount of work that is generally well performed. I think my key clarification really revolves around a comparison of these data to the DepMap data.

POINT-BY-POINT RESPONSE TO REVIEWER COMMENTS

Reviewer #1 (Remarks to the Author):

The paper by Houles et al finds CDK12 to be a downstream target of RAS/MAPK signalling pathway, that is largely responsible for drug resistance during melanoma treatment. The authors identify threonine 548 (Thr548) in CDK12 proline-rich region as a substrate of ERK1/2 kinases and propose this signalling to be responsible for CDK12 hyperactivation in BRAF-mutated melanoma cells. Using covalent CDK12/CDK13 inhibitor THZ531 they show that treatment of BRAF-mutated melanoma cells results in downregulation of long genes (particularly DNA-repair ones) and upregulation of short genes including many components of pro-growth AP-1 and NF-KB pathways. Finally, they show that inhibition of these two pathways synergizes with CDK12 inhibition and results in growth suppression of melanoma cells. The paper shows important findings, particularly potential link between RAS/MAPK pathway and CDK12 and synthetic lethality between CDK12 and AP-1/NF-KB pathways will be of a broad scientific interest. I would support publication in Nature Communications after the following concerns are addressed.

Reply: We thank the reviewer for the positive evaluation of our work. We did our best to address all concerns as shown below.

Major comments

1) Fig. 3: It is not clear why P-Ser7 is phosphorylated in the GST-CTD purified from bacteria (Fig. 3 b, e, g, Supp. Fig. 1a). Proteins expressed in bacteria are not phosphorylated. This would also explain why the P-Ser7 band is not sensitive to the THZ531 treatment (Supp. Fig. 1). Is the P-Ser7 band phosphatase-sensitive? Thus, the results of the experiment can not be used to conclude that CDK12 phosphorylates Ser2 and Ser5, but not Ser7. There is a relatively strong evidence that CDK12 is a promiscuous CTD kinase in human cells. This conclusion is supported by various in vitro kinase assays (1,2) and by selective inhibition of CDK12 in cell lines carrying analog-sensitive CDK12 alleles (3-6). These papers show small and distinct effects of CDK12 on the CTD phosphorylation and the results should be considered when roles of CDK12 in the CTD phosphorylation and transcription are introduced. Another issue in the manuscript are possible off-target effects of higher concentration of THZ531 (>100 nM) (7). Lower concentrations of THZ531 would be better to secure selective inhibition of CDK12 (see also comment in points 7 and 8). All these issues are reviewed for example here (8) and should be considered in experimental design and data interpretation.

Reply: We thank the reviewer for commenting on our results involving Ser7 phosphorylation. As requested, we determined if the phospho-Ser7 band was sensitive to phosphatase treatment. Much to our surprise, we found that the phospho-Ser7 band was in fact phosphatase-insensitive. As these results suggest that the phospho-Ser7 antibody recognizes unphosphorylated RPB1, we have decided to remove all data pertaining to the use of this antibody. This does not change any of our conclusions, as we have quantified CDK12 activity based on its ability to phosphorylate RPB1 at Ser2 and Ser5, which are known to be phosphorylated by CDK12^{1,2}. We

also updated the manuscript to indicate that CDK12 may regulate gene expression in several ways, and not only through the regulation of Pol II-dependent transcription, as indicated.

The second point is about possible off-target effects of THZ531 when used at concentrations above 100 nM, which we have addressed in multiple ways in two BRAF-mutated melanoma cell lines. First, we have validated the effect of THZ531 on gene expression using a range of concentrations (100 to 500 nM), as shown by western blotting and qPCR (see **Fig. 4h, 4i, and Supplementary Fig. S2d, e; S3c, f, h**). Second, we used RNAi to demonstrate that specific depletion of CDK12 results in similar changes in gene expression (see **new Supplementary Fig. S3g**). Third, we compared the effect of THZ531 with another CDK12 inhibitor, SR-4835, and found that both molecules affect gene expression in a similar manner (see **new Supplementary Fig. S3d, e**). Combined with the fact that most of our synergy experiments were performed using THZ531 at 100 nM (**Fig. 5 and 6**), and that many studies use this compound at concentrations between 100 and 400 nM³⁻⁵, we believe that our data are strongly suggestive of CDK12-dependent effects.

2) Fig. 3b, e: If change in in vitro activity is a result of (or correlates with) phosphorylation of Thr548 this needs to be presented on the western blots. This is an important control to support conclusion of the paper.

Reply: We thank the reviewer for this comment. We agree that showing Thr548 phosphorylation in Figs. 3b and 3e would nicely complement the kinase assays. However, the antibody we generated against phospho-Thr548 is not very efficient and only detects phosphorylated CDK12 when large amounts are immunoprecipitated (**as in Fig. 3e**, using immunoprecipitated Myc-tagged CDK12). As this antibody is not sensitive enough to detect endogenous CDK12 phosphorylation, and there is no commercially available phospho-antibodies against phospho-Thr548, we could not include requested results in the paper.

3) Fig. 3b, lane 2 and Supp. Fig. 1a, lane 3: Discrepancy between the figures - “not activated“ CDK12 shows strong phosphorylation of Ser2 in Supp. Fig.1a, but no phosphorylation in Fig.3b.

Reply: The main difference between experiments shown in Supplementary Fig. 1a (**now Supplementary Fig. 2a**) and Fig. 3b (**now Fig. 3a**) is that serum-growing cells were used to validate CDK12 *in vitro* kinase assays in the former. Conversely, the kinase assay shown in **Fig. 3a** was performed using serum-starved cells, which attenuates basal ERK1/2 activity (see ERK1/2 phosphorylation levels) and lowers basal CDK12 activity. In addition, exposure times are different between both figures: whereas basal CDK12 activity required longer exposures (**as shown in Supplementary Fig. S2a**), shorter exposures were used to better encapsulate the increase in CDK12 activity due to PMA stimulation in **Fig. 3a**). We hope that these clarifications help support our conclusions that PMA increases CDK12 activity.

4) Fig.3c: Can loading controls other than RBP1 and ERK1/2 be shown? It is hard to compare between cell lines because the RBP1 levels differ significantly. The P-Ser5 and P-Ser7 levels need to be also shown (see point 1). In addition, what are the levels of total CDK12 protein and P-Thr548 in these cell lines? If P-Ser2 increase (or possibly P-CTD increase?) is due to the hyperactivated CDK12 (line 174 and title of the paper) this should be documented by a corresponding increase in P-Thr548.

Reply: We thank the reviewer for this comment. We have now added other loading controls, including CDK12 and Tubulin, which show nearly equal protein loading between lanes (see **Supplementary Fig. S3b**). We have added phospho-Ser5, but not phospho-Ser7, for reasons indicated above. Finally, for reasons also indicated above, the phospho-Thr548 signal on endogenous CDK12 could not be added to the panel. Nevertheless, these new results show that phospho-RPB1 (Ser2 and Ser5) levels are higher in BRAF-mutated melanoma cell lines compared to melanocytes, which was the original goal of the experiment.

5) Does ERK1/2 activation/inhibition affect the global state of the CTD phosphorylation (P-Ser2,5,7)? This should be tested in total cell lysates of selected melanoma cell lines.

Reply: To answer this comment, we now provide new experimental findings showing that inhibition of MEK1/2 and ERK1/2 activity results in reduced RPB1 phosphorylation at both Ser2 and Ser5 in A375 and Colo829 cell lines (**Fig. 3d; Supplementary Fig. S2c**).

6) I would suggest to use expression of well-characterized CDK12-dependent DNA-repair/replication genes as a readout of CDK12 activity after ERK1/2 inhibition/activation to make a stronger case for the role of ERK1/2 in CDK12 activation. This should give much clearer picture than solely relying on in vitro kinase assays with the CTD (with all the caveats described above).

Reply: We thank the reviewer for this comment, which allowed us to better characterize the link between ERK1/2 and CDK12 activity. We now provide new experimental findings showing, as expected and predicted by RNA-seq, that CDK12 inhibition (using THZ531 and SR-4835) decreases the levels of genes involved in the DNA damage response in melanoma cell lines (**Supplementary Fig. S2d, e; S3c, d**). Concurrently, ERK1/2 inhibition (using BVD-523) was found to decrease the same set of genes in A375 and Colo829 (**Supplementary Fig. S2f, g**). Our results, together with RPB1 phosphorylation, reinforce the idea that ERK1/2 regulate CDK12 activity.

7) Fig. 4: RNA-seq was performed with relatively high concentrations of THZ531 (500 nM). To exclude a possibility of upregulation of genes in AP-1 and NF- κ B pathways due to off-target effects, RT-qPCR with lower concentrations of THZ531 and/or another CDK12 inhibitor SR-48359 should be performed. Increase in protein levels of the components of AP-1 and NF- κ B pathways (Fos, Jun, Gadd45a or others) should be documented by western blotting.

Reply: As indicated above, we have validated the effect of THZ531 on gene expression using a range of concentrations. Indeed, using western blotting and qPCR, our new data show that THZ531 affects gene expression in a dose-dependent manner (100 to 500 nM) (see **new Fig. 4h; Supplementary Fig. S3c**). In addition, we have used RNAi to demonstrate that the specific depletion of CDK12 results in similar changes in gene expression (see **new Supplementary Fig. S3g**). As requested, we have also tested SR-4835, which is another CDK12 inhibitor, and found that it affects gene expression in a similar manner to THZ531 (see **new Supplementary Fig. S3d, e**). Thus, our results support the idea that THZ531 upregulates the AP-1 and NF- κ B pathways through the specific inhibition of CDK12.

8) Fig. 7: THZ1 is primarily a CDK7 inhibitor with some degree of affinity towards CDK12 and

CDK13 (10). At minimum, this should be clearly stated in the text. CDK7 activates many other kinases, particularly CDK9, CDK1, CDK2, CDK4, CDK6 (11,12) and this can also significantly affect the results. Did authors consider using other CDK12 inhibitor such SR-48359 if THZ531 can't be used in in vivo studies? This compound was used in vivo models (9).

Reply: We thank the reviewer for this precision, which was added to the manuscript. With regards to SR-4835, which is another CDK12 inhibitor, we have performed several additional experiments that were added to the manuscript (see **Fig. 4; Supplementary Fig. S3**). At this point, we did not see fit to redo all *in vivo* experiments with this compound. More work will be required to fully characterize the mechanism of action of SR-4835, as it appears to be distinct from THZ531 and THZ1 ⁶.

9) Discussion, lanes 292, 320: Additional experiments (see above) are required to confirm the hyperactivated state of CDK12 in melanoma.

Reply: This comment relates to **point #3**, which challenges the idea that CDK12 is more active in BRAF-mutated melanoma cells. As indicated above, it is important to note that CDK12 activity appears to follow that of the RAS/MAPK pathway. Our results demonstrate that CDK12 activity is dependent on the RAS/MAPK pathway in HEK293 cells (**Fig. 3a**), but also in melanoma cells (**Fig. 3c**). Based on kinase assays using an unphosphorylatable mutant of CDK12 (T548A), we show that ERK1/2-mediated phosphorylation of CDK12 promotes its activity (**Fig. 3f**). Together, these experiments suggest that CDK12 is more active in BRAF-mediated melanoma cells.

Minor comments

1) Fig. 1b and line 99: the GFP biotinylation signal seems to be stronger in comparison to ERK1/2.

Reply: We have modified the text to put less emphasis on the similarities/differences between protein biotinylation patterns. We think that the higher biotinylation signal associated with BirA-actually makes it a better control, as all potential hits were normalized to this bait.

2) Fig. 1: It is not clear if HEK293 empty vector (EV) or GFP-BirA cells (or both) were used as a control for the MS data analysis.

Reply: We thank the reviewer for pointing out that the manuscript did not clearly indicate which control was used for BioID analysis. As is now indicated in the text, it is in fact BirA-GFP cells that were used as control, and not cells expressing an empty vector.

3) Line 121: It would be informative to show figure documenting conservation of Thr548 in CDK12 across species.

Reply: As requested, we have added an alignment of CDK12 sequences from different species (**Fig. 2g**).

4) Fig. 2: EGF abbreviations should be spelled out, use of serum starvation experiment should be briefly explained as well as meaning of phospho-markers (P-ERK1/2, (P)-rpS6, P-Akt-S473...).

Anti-pS/T-P antibody is not described in the materials and methods. They should also explain why they use overexpressed CDK12 and not the endogenous one to show phosphorylation of Thr548.

Reply: We made the appropriate change in the manuscript. Importantly, and as indicated above, we now explain why it is not possible to show Thr548 phosphorylation on endogenous CDK12.

5) Fig. 4h: concentrations of THZ531 should be specified.

Reply: We thank the reviewer for noticing this omission. The figure legend now indicates the concentrations of THZ531 used.

6) Fig. 7b: 2nd graph – replace MLN120B with B1601245.

Reply: We thank the reviewer for noticing this mistake. We made the appropriate change.

7) Methods:

Lane 349: Correct HEK293 BirA-ERK1/ERK2; A375 cells stably expressing Myc-CDK12(WT/Thr548Ala) should be also described.

Lane 368: Correct to “Stable Inducible Cell Line”

Lane 440: Delete section “Immunofluorescence Microscopy” (not used in the paper).

Lane 507: Delete Immunohistochemistry– Caspase-3 antibody (not used in the paper).

Lane 527 : Correct “with immunoprecipitated CDK12 as kinase”

Reply: We have made all requested corrections and more.

References:

1. Bosken, C.A. et al. The structure and substrate specificity of human Cdk12/Cyclin K. *Nat Commun* 5, 3505 (2014).
2. Bartkowiak, B. & Greenleaf, A.L. Expression, purification, and identification of associated proteins of the full-length hCDK12/CyclinK complex. *J Biol Chem* 290, 1786-95 (2015).
3. Chirackal Manavalan, A.P. et al. CDK12 controls G1/S progression by regulating RNAPII processivity at core DNA replication genes. *EMBO Rep* 20, e47592 (2019).
4. Tellier, M. et al. CDK12 globally stimulates RNA polymerase II transcription elongation and carboxyl-terminal domain phosphorylation. *Nucleic Acids Res* 48, 7712-7727 (2020).
5. Bartkowiak, B., Yan, C. & Greenleaf, A.L. Engineering an analog-sensitive CDK12 cell line using CRISPR/Cas. *Biochim Biophys Acta* 1849, 1179-87 (2015).
6. Fan, Z. et al. CDK13 cooperates with CDK12 to control global RNA polymerase II processivity. *Sci Adv* 6(2020).
7. Zhang, T. et al. Covalent targeting of remote cysteine residues to develop CDK12 and CDK13 inhibitors. *Nat Chem Biol* 12, 876-84 (2016).
8. Pilarova, K., Herudek, J. & Blazek, D. CDK12: cellular functions and therapeutic potential of versatile player in cancer. *NAR Cancer* 2, zcaa003 (2020).
9. Quereda, V. et al. Therapeutic Targeting of CDK12/CDK13 in Triple-Negative Breast Cancer. *Cancer Cell* (2019).
10. Kwiatkowski, N. et al. Targeting transcription regulation in cancer with a covalent CDK7 inhibitor. *Nature* 511, 616-20 (2014).
11. Larochelle, S. et al. Cyclin-dependent kinase control of the initiation-to-elongation switch of RNA polymerase II. *Nat Struct Mol Biol* 19, 1108-15 (2012).
12. Schachter, M.M. et al. A Cdk7-Cdk4 T-loop phosphorylation cascade promotes G1 progression. *Mol Cell* 50, 250-60 (2013).

Reply: Many of the above-cited papers were included in the manuscript.

Reviewer #2 (Remarks to the Author):

Houles et al performed proximity labeling using ectopic BirA-ERK1 and 2, cross referenced their results with two other published datasets and came up with a list of 23 proteins of interest, of which they chose to focus on CDK12. The authors nicely demonstrate by multiple means that ERK1/2 phosphorylate CDK12 on Thr548, including going so far as to generate their own phospho-specific antibody for this site and validating this phosphorylation by epistatic analysis. Nice work! The authors then provide evidence that PMA-induced phosphorylation of CDK12 promotes its kinase activity based on an IP-kinase assay using the known substrate, the CTD domain of a pol II subunit. Again, these are nice experiments. To tie this into a function of CDK12, the authors performed RNAseq on two cell lines with high MAPK activation, plus and minus a CDK12 inhibitor. This led to a series of drug synergy experiments with JNK and IKKbeta inhibitors, which is fine and adds some clinical relevance to the study, but in terms of mechanism is underdeveloped and somewhat over-interpreted. The authors then take it home with an in vivo study, showing some decrease in tumor growth when a CDK12 inhibitor is combined with an IKKbeta inhibitor. All in all, this is a great study, especially the first half. I also appreciate that the authors add clinical relevance to their work by performing RNAseq analysis to find a possible downstream druggable target to combine with a CDK12 inhibitor. Concerns are all addressable. Recommend the authors have the opportunity to revise their manuscript for resubmission.

Reply: We thank the reviewer for the positive and enthusiastic evaluation of our work. All concerns were addressed as shown below.

Major concerns

1) Key to the success of BirA proximity labeling is validation that the fusion protein actually functions correctly. Please confirm that BirA-ERK1 reverts a null phenotype (e.g. restores proliferation, restores transcriptional signatures as assessed by whole genome RNAseq analysis, restores the phosphorylation of ERK substrates by whole genome phosphor-proteomics, or whatever comprehensive phenotype the authors choose) of ERK1 sgRNA-treated cells validated to lack ERK protein expression, please include the negative control (BirA alone) and positive control (unfused ERK1) expression constructs. Repeat the same experiment with BirA-ERK2. Please note that this is a critical requirement for publication, and thus the authors are strongly encouraged to perform these experiments.

Reply: To ensure that BirA-ERK1/2 fusion proteins function correctly, we executed a series of experiments as follows. **First**, we determined if fusion proteins could immunoprecipitate with RSK2, which is a well-known ERK1/2-binding protein and substrate^{7,8}. We found that, compared to BirA alone and BirA-GFP, both BirA-ERK1 and BirA-ERK2 could readily immunoprecipitate HA-tagged RSK2 expressed in HEK293 cells (see **Supplementary Fig. S1b**), indicating that ERK1/2-fusion proteins retain the ability to interact with downstream substrates. **Second**, we depleted ERK1/2 in HEK293 cells, and determined whether ectopic expression of BirA-ERK1 or BirA-ERK2 could restore downstream signaling events. We found that both fusion proteins restored RSK1 and S6 phosphorylation in depleted cells, as also found with HA-tagged ERK1 (see **Supplementary Fig. S1c**), indicating that both fusion proteins retain the ability to phosphorylate downstream substrates. While we did not use CRISPR-related approaches, we hope that these

additional experiments, which include RNAi-mediated loss-of-function, provide sufficient evidence for the functionality of the BirA-ERK1/2 fusion proteins. We thank the reviewer for suggesting these important control experiments, which help increase the significance of our BioID work.

2) The RNAseq analysis is somewhat over-interpreted, as this is really analysis of a CDK12 inhibitor (plus no data was provided that the inhibitor works in the cell assayed) and not CDK12 activity per se. What I mean by this is would you expect the same profile in cells with low CDK12 phosphorylation? Is this a dataset of CDK12 function in general, rather than an activated response? Either rephrase the interpretation of this experiment or cross reference it with treatment of melanoma line without high CDK12 phosphorylation with the CDK12 inhibitor, or if none exist, than some other cell line with low CDK12 phosphorylation. Again, the point here is whether the authors are looking at the effect of high CDK12 activity due to robust MAPK activation, or the effect of CDK12 function in general.

Reply: We understand the point the reviewer is trying to make, and we have addressed the specificity of the effects observed in multiple ways. Using qPCR to monitor the expression of DNA repair genes (**Supplementary Fig. S2d, e**), or western blotting to determine RPB1 phosphorylation (**Fig. 3d; Supplementary Fig. S2c**), we show that CDK12 inhibition disrupts known CDK12-dependent functions. We corroborated these data using a second CDK12 inhibitor (SR-4835) or RNAi, and observed similar changes in gene expression in response to CDK12 depletion (see **Fig. 4; Supplementary Fig. S3**). Together, these new data provide additional evidence showing that the CDK12 inhibitors function as expected. The reviewer also wonders if the CDK12-dependent transcriptome of melanoma cells reflects basal CDK12 function or an activated response. This is a difficult question to answer, as most melanoma cell lines harbor mutations that hyperactivate the RAS/MAPK pathway. We think that melanoma cells harbor an exacerbated CDK12 response, which may amplify normal CDK12 functions.

3) Please consider softening the conclusions on the synergy experiments, as classic epistatic analysis was not performed to validate the mechanism of this synergy. While I have no problem with the idea of the RNAseq analysis supporting drug synergy experiments, I would caution the authors on over-interpreting these in terms of a direct connection to CDK12 function, as one could imagine grossly disrupting the mRNA pool with a CDK12 inhibitor could set up all kinds of potential synthetic lethality, and JNK and IKKbeta are pretty broad in their effects.

Reply: We understand the point from the reviewer, as we have not performed classic epistatic experiments to validate/confirm the exact mechanisms involved in the different synergies. We therefore softened some of our conclusions related to synergistic effects.

4) The authors' line of investigation in figure 7 suggests that CDK12 may offer a new druggable kinase to treat melanoma, indeed this was the impetus for the study as described in the introduction, yet the effect on xenograft growth appears rather modest. As a key control to put this work into context with current therapies, please repeat Fig 7d,e, but instead test with vemurafenib (or

dabrafenib) AND trametinib. This is a really important control, as it will help inform on the possible utility of this new drug synergy in the clinic.

Reply: While our study indeed suggests the identification of new druggable targets in melanoma, this work remains quite preliminary compared to the use of optimized drugs such as vemurafenib and trametinib. Indeed, both of these FDA-approved drugs have been thoroughly characterized in pre-clinical models, including A375 xenografts⁹⁻¹⁵, and very little additional information would be gained from repeating this work. We believe that a fair and clinically informative comparison would require using optimized and clinically tested or approved compounds, which are not available for CDK12. Moreover, our study does not pretend to having found replacements for these FDA-approved drugs, but rather that our results show a good potential that deserves future development. Thus, much more remains to be done to optimize these new drug synergies *in vitro* and using pre-clinical models of melanoma.

5) Please test whether the Phospho-Thr548 antibody detects high CDK12 phosphorylation on clinical melanoma samples, as this would go a long way to supporting the contention that CDK12 is upregulated during melanomagenesis. A small panel of tumors is fine, as the study is already quite comprehensive.

Reply: We understand that it would be interesting to determine Thr548 phosphorylation in clinical samples, but as mentioned above, we are limited by our “in-house” phosphospecific antibody. As shown below, the antibody is not efficient enough to detect endogenous CDK12 protein phosphorylated at Thr548.

Minor concerns

-Please move Fig 1a to the supplement, as at this point proximity labeling is a well-known assay

Reply: We understand the reviewer, but feel that the data is more easily understood with a small schematic summarizing the method. Reviewer 3 actually requested that we improved this panel, which we have done in the revised manuscript.

-Please clarify that the BirA-ERK fusion proteins are ectopically expressed, and please compare the level of these proteins to the endogenous ones. There is no problem with using ectopic BirA-ERK, others have done this with great success with other BirA fusion proteins, but it is helpful to know the expression level, especially in terms of comparing the interactome to other datasets.

Reply: We thank the reviewer for noticing this error in the manuscript, which now more clearly describes how the baits were expressed for BioID analysis. We also added a panel showing the expression level of BirA-ERK1/2 fusion proteins compared to endogenous proteins (see **Supplementary Fig. S1a**).

-please either normalize the BirA-ERK dataset against one generated with BirA alone (in the presence of biotin), or note in the results that the dataset was not normalized to BirA alone, as this is a common control for these types of experiments.

Reply: We apologize for the lack of clarity. We added a few sentences in the Results section and in the Methods section indicating that the BirA-ERK1/2 datasets were normalized to that of BirA-GFP, which is an appropriate control.

-the authors claim that serum does not lead to CDK12 phosphorylation because it activates the PI3k/AKT pathway, yet ERK is clearly phosphorylated, please consider revising the text to better reflect the results, for example, this result suggests that potent MAPK activation leads to CDK12 phosphorylation.

Reply: We have made the requested changes to the manuscript.

-please include the full-length gels in the supplement, this is particularly important to gauge how specific the antibodies are, for example, for the new Phospho-Thr548 antibody.

Reply: We understand the reviewer and, as requested, we have added full-length gels for all western blots present in the manuscript in the Source Data file. However, and as mentioned above, the phospho-antibody generated against CDK12 (pT548) is not very efficient and only detects phosphorylated CDK12 when large amounts are immunoprecipitated.

-please explicitly state in the figure legends the number of replicates and type (technical or biological) for each experiment, and when multiple experiments have been performed, please consider reporting the averages in a graph in the supplemental data. I leave this to the authors to decide which experiments to quantitate, but minimally all replicates need to be included in the supplemental data as full-length gels and so forth. This simply provides one metric to help gauge the degree of rigor for each experiment.

Reply: We understand the reviewer and all information concerning replicates (number, type, value) have been added to the legend or the supplementary data as instructed by the editorial board.

-Is there an active ERK mutant the authors could use to validate phosphorylation on Thr548 in cells, analogous to the experiment in Fig. 2g? Such an experiment would further strengthen their claims.

Reply: MEK1/2 are very specific protein kinases, as their only known substrates are ERK1 and ERK2. For this reason, we have used an activated form of MEK1 (S218/222D) that specifically activates ERK1 and ERK2. These experiments revealed that ERK1/2 activation is sufficient for CDK12 phosphorylation (**Fig. 2h**).

-Fig 3a is not necessary, these are pretty standard approaches, so please move to the supplement.

Reply: We removed the schematic altogether, as we felt that it was even less useful in the supplement.

-T>D mutants can sometimes act as phosphomimics, please generate a Thr548>Asp CDK12 mutant and determine whether it has increased kinase activity in vitro and in cells. If positive, this would clearly be nice supporting data, if negative, the authors have ruled out an obvious independent test of their hypothesis.

Reply: We thank the reviewer for this point. We have generated a potential CDK12 phosphomimic by mutation of Thr548 to Glu, but this mutation did not increase CDK12 kinase activity *in vitro*.

-Fig 7c really is not required, please move to supplement.

Reply: While we agree that this panel is not absolutely necessary, we feel that it provides key information to rapidly understand how the experiment was performed. For this reason, we feel that the panel should remain in Fig. 7.

-please speculate on the relatively poor overlap between proximity labeling and the other approaches described in Fig 1f in the discussion.

Reply: We have included a brief statement in the discussion that describes the overlap (31%) between our BioID dataset and the ERK1/2 compendium. We feel that this level of overlap is not that low, considering the fact that BioID identifies proximity interactors and not specifically substrates. In addition, the overlap increases to 40% when comparing our data to Scansite predictions, which suggest that we have identified many new potential ERK1/2 substrates.

-various typos in the manuscript, especially the materials and methods that need to be corrected.

Reply: We thank the reviewer for this remark. We proofread the article and corrected various typos.

Reviewer #3 (Remarks to the Author):

In their paper Houles and co-workers explore the role of CDK12 in BRAF mutant melanoma. This is generally a well written paper but I do have several suggestions/recommendations below:

1. What is a “prime” synthetic lethal target? The word “prime” tells me nothing about the role of CDK12 in the fitness of BRAF-mutant melanoma cells. I’d suggest re-working the title.

Reply: We simply removed the word “prime” from the title, which actually improved its clarity. We thank the reviewer for this suggestion.

2. I think Figure 1A could be improved. Even with my big screen I can’t see all of the writing so I can’t imagine it will be visible when printed. Maybe this part of the figure could be enlarged and moved to the supplementary data.

Reply: We thank the reviewer for this comment. We have enlarged panel 1A to more clearly show the BioID procedure.

3. Error in figure legend 1c should read 0.918 not 0918.

Reply: We made the appropriate change.

4. In figure 1e – I can see that ERK1/2 interactors are grouped. Does this grouping have any statistical significance? The suggestion is that there is an enrichment for factors involved in (for example) transcription, cell trafficking etc but that’s not what this figure tells me. Could this be clarified please?

Reply: We thank the reviewer for this comment. The manuscript now includes the *p*-values for each of the significant enrichments displayed in Figure 1e.

5. I like the way the authors have used multiple approaches to end up with 31 interactors. This is a good strong approach. Were there any established interactors that were missed? Can they give the readers a sense for the false negative rate?

Reply: ERK1/2 are known to interact with many adaptor proteins and substrates in cells. From our data, we think the more important number is 56, which is the overlap between our BioID dataset (179) and the ERK1/2 compendium. According to this number, 31% of the proteins we identified by BioID were also found to be phosphorylated in an ERK1/2-dependent manner. While our approach did not identify a large number of known ERK1/2-interacting proteins, we were relatively pleased to see that one third of identified proteins were likely to be ERK1/2 substrates. This number was refined even more by determining the presence of ERK1/2-docking motifs, which was found in the resulting 31 proteins. The manuscript now presents these important numbers in a clearer way.

6. Line 118 – I would say “23 novel potential ERK1/2 substrates....”

Reply: We made the appropriate change.

7. As a melanoma biologist/geneticist I got a bit lost when THZ531 was introduced. Could you please explain what it is and why you reasoned that it would abrogate CDK12-mediated phosphorylation? I found the introduction of this compound somewhat abrupt.

Reply: We added a sentence introducing THZ531 upon its first appearance in the text of the manuscript.

8. As an aside the other place where the MAPK pathway is activated via BRAF mutation is in paediatric low grade glioma.

Reply: We thank the reviewer for this information. We will certainly look into this.

9. I wonder if the authors could explain why CDK12 is not an essential gene in the all of the BRAF mutant melanoma cell lines CRISPR screened by DepMap: <https://depmap.org/portal/gene/CDK12?tab=overview> or alternatively use these data to further explore the role of CDK12 in this disease?

Reply: We were surprised by this comment, as *CDK12* is in fact listed as a “common essential” gene in DepMap. We verified that melanoma cell lines were no exception, and indeed, we found that nearly all melanoma cell lines tested as part of the DepMap initiative have a similar dependency towards *CDK12* (see summary below, taken directly from the DepMap web site).

10. What happens to the expression of long and short genes when cells are sick? I guess I wonder if the correlations reported (e.g. gene length relative to CDK12 activity) are really a read out of cell fitness rather than being a phenotype (selectively) associated with perturbation of CDK12.

Reply: While we understand the reviewer's concern, we would like to indicate that we have analyzed the transcriptome of melanoma cells following very short inhibitor treatments (6 h). In this short timeframe, we have not observed any issues with cell survival, suggesting that changes in gene expression are likely the result of target inhibition. In fact, our data indicate that issues with cell viability occur much later (i.e., after a minimum of 24h at the doses used). In addition, we have observed similar defects in gene expression upon CDK12 depletion (see **Supplementary Fig. S3g**), indicating again the probable specificity of these effects. In addition, the manuscript now contains data from experiments using SR-4835, another CDK12 inhibitor (**Supplementary Fig. S3d, e**).

11. Figure 5 b,c. Fig 6bc and fig7b are (I'm sorry) a bit meaningless. I really don't understand the message of these figures. They all look the same to me.

Reply: We have modified these panels to better show the range of doses required to demonstrate significant synergy and Bliss score positivity. These panels are necessary to show that Bliss positivity occur at multiple doses of both compounds.

12. Figure 7f needs a scale bar.

Reply: We made the appropriate change.

Overall this paper includes an enormous amount of work that is generally well performed. I think my key clarification really revolves around a comparison of these data to the DepMap data.

Reply: We thank the reviewer for their appreciation of our findings, and for boiling down their comments. As indicated above, CDK12 is indeed an essential gene in melanoma cells, and thus we feel that this issue is now resolved. A mention to that effect was also added to the manuscript.

REFERENCES

- 1 Bartkowiak, B. & Greenleaf, A. L. Expression, purification, and identification of associated proteins of the full-length hCDK12/CyclinK complex. *The Journal of biological chemistry* **290**, 1786-1795, doi:10.1074/jbc.M114.612226 (2015).
- 2 Bosken, C. A. *et al.* The structure and substrate specificity of human Cdk12/Cyclin K. *Nature communications* **5**, 3505, doi:10.1038/ncomms4505 (2014).
- 3 Krajewska, M. *et al.* CDK12 loss in cancer cells affects DNA damage response genes through premature cleavage and polyadenylation. *Nature communications* **10**, 1757, doi:10.1038/s41467-019-09703-y (2019).
- 4 Shyamsunder, P. *et al.* THZ531 Induces a State of BRCAness in Multiple Myeloma Cells: Synthetic Lethality with Combination Treatment of THZ 531 with DNA Repair Inhibitors. *International journal of molecular sciences* **23**, doi:10.3390/ijms23031207 (2022).
- 5 Bayles, I. *et al.* Ex vivo screen identifies CDK12 as a metastatic vulnerability in osteosarcoma. *The Journal of clinical investigation* **129**, 4377-4392, doi:10.1172/JCI127718 (2019).

- 6 Dieter, S. M. *et al.* Degradation of CCNK/CDK12 is a druggable vulnerability of colorectal cancer. *Cell reports* **36**, 109394, doi:10.1016/j.celrep.2021.109394 (2021).
- 7 Smith, J. A., Poteet-Smith, C. E., Malarkey, K. & Sturgill, T. W. Identification of an extracellular signal-regulated kinase (ERK) docking site in ribosomal S6 kinase, a sequence critical for activation by ERK in vivo. *The Journal of biological chemistry* **274**, 2893-2898, doi:10.1074/jbc.274.5.2893 (1999).
- 8 Dalby, K. N., Morrice, N., Caudwell, F. B., Avruch, J. & Cohen, P. Identification of regulatory phosphorylation sites in mitogen-activated protein kinase (MAPK)-activated protein kinase-1 α /p90rsk that are inducible by MAPK. *The Journal of biological chemistry* **273**, 1496-1505, doi:10.1074/jbc.273.3.1496 (1998).
- 9 Gilmartin, A. G. *et al.* GSK1120212 (JTP-74057) is an inhibitor of MEK activity and activation with favorable pharmacokinetic properties for sustained in vivo pathway inhibition. *Clinical cancer research : an official journal of the American Association for Cancer Research* **17**, 989-1000, doi:10.1158/1078-0432.CCR-10-2200 (2011).
- 10 Potu, H. *et al.* Downregulation of SOX2 by inhibition of Usp9X induces apoptosis in melanoma. *Oncotarget* **12**, 160-172, doi:10.18632/oncotarget.27869 (2021).
- 11 Sun, C. *et al.* Reversible and adaptive resistance to BRAF(V600E) inhibition in melanoma. *Nature* **508**, 118-122, doi:10.1038/nature13121 (2014).
- 12 Venkatesan, A. M. *et al.* Ligand-activated BMP signaling inhibits cell differentiation and death to promote melanoma. *The Journal of clinical investigation* **128**, 294-308, doi:10.1172/JCI92513 (2018).
- 13 Waizenegger, I. C. *et al.* A Novel RAF Kinase Inhibitor with DFG-Out-Binding Mode: High Efficacy in BRAF-Mutant Tumor Xenograft Models in the Absence of Normal Tissue Hyperproliferation. *Molecular cancer therapeutics* **15**, 354-365, doi:10.1158/1535-7163.MCT-15-0617 (2016).
- 14 Wang, L. *et al.* An Acquired Vulnerability of Drug-Resistant Melanoma with Therapeutic Potential. *Cell* **173**, 1413-1425 e1414, doi:10.1016/j.cell.2018.04.012 (2018).
- 15 Yadav, V. *et al.* The CDK4/6 inhibitor LY2835219 overcomes vemurafenib resistance resulting from MAPK reactivation and cyclin D1 upregulation. *Molecular cancer therapeutics* **13**, 2253-2263, doi:10.1158/1535-7163.MCT-14-0257 (2014).

End of the response to reviewers' comments.

Reviewers' Comments:

Reviewer #1:

Remarks to the Author:

The authors addressed most of my comments and significantly improved the manuscript. I understand some of the technical difficulties, but I still have three requirements that need to be addressed before the acceptance for publication:

1) The home-made P-Thr548 CDK12 antibody is the key reagent linking specific changes in phosphorylation of CDK12 to activated RAS/MAPK pathway and thus the antibody needs to be properly characterized. As stated by the authors, the antibody does not recognize endogenous CDK12 phosphorylated on Thr548 and the western blot presented in the response to the comment 5 of the reviewer 2 shows a lot of bands and none of them appears to be sensitive to the CDK12 knockdowns. To rigorously confirm the specificity of the antibody, the authors need to show that the P-Thr548 CDK12 band in overexpressed and immunoprecipitated CDK12 (Fig. 2i, 3e) is indeed sensitive to phosphatase treatment and CDK12 depletion. It should be also directly stated in the result section that P-Thr548 CDK12 antibody does not recognize the endogenous protein, only overexpressed and immunoprecipitated one (as shown in Figs 2i, 3e).

2) I noted that western blots in Figs. 2i, 3e, 3d and in many other figures in the manuscript are disproportionately cropped (please compare to original western blots presented in the source data file). The incorrect manipulation with the original gels distorts meaning of loading controls, effects of treatments on phosphorylation etc. This is not acceptable and needs to be corrected throughout the manuscript.

3) I appreciate the authors mention that THZ1 is in fact CDK7 inhibitor (as response to my comment no 8). The added references 44 and 45 are, however, misleading (line 301) as only the reference 46 shows that the THZ1 also partially inhibits CDK12 and CDK13. Given that CDK7 inhibition globally downregulates transcription (ref. 46) and the kinase affects activity of many other CDKs (ref. 44, 45), the limitations of this in vivo experiment need to be clearly stated/discussed in the discussion as no selective CDK12 inhibitor was in fact used for in vivo studies (i.e. the observed outcome can be completely independent of the CDK12 function).

Reviewer #2:

Remarks to the Author:

The authors have address the previous concerns (or opted to not make some changes). One remaining comment that is easily addressed with a simple reference to the literature remains that of Major comment #4. Specifically, the authors argue that as others have already tested dabrafenib plus trametinib, there is little value in comparing their drug combination to the clinical one in a xenograft model. Fair point. However, related to the importance of putting their findings into context with current clinical practice, it would be ever-so-valuable to make a compare the response the authors document in this study with the aforementioned published studies. With the exception of this last easily addressed point, the authors were highly responsive.

In terms of other previous concerns, I leave it to the authors to decide if they would like to further revise their manuscript in regards to:

Major comment #5- The authors clearly demonstrate that their phospho-specific antibody cannot detect endogenous CDK12 phosphorylation, thank you. Related to this, perhaps the authors could clarify when endogenous versus ectopic proteins are being analyzed in the text. While to the authors' credit, these details can be found in the methods or legends, having it in the text really helps navigate the experiments.

Minor comment #7- The authors make the case that because ERK is the only substrate for MEK that a gain-of-function mutant of ERK is not required. I note here for the purpose of information that there are both ERK activation mutants and reports of MEK phosphorylating other proteins

(although admittedly very few and it is true that the dogma is that MEK only phosphorylates ERK).

Minor comment #9- To their credit, the authors generated and then tested a Thr548>Asp mutant of CDK12, finding it did not increase activity. Please consider including this as a supplemental figure, as your hard work could save someone the frustration of trying this experiment. Moreover, as it is such an obvious epistatic experiment, please consider noting that the mutant does not increase activity in the text.

Reviewer #3:

Remarks to the Author:

Generally I think the authors have done a good job of responding to my comments. Regarding point 9 relating to DepMap the authors should look at the scores for essential melanoma genes - for example SOX10 or BRAF in BRAF mutant lines. In both cases the genes are strongly essential (with Chronos scores less than -1). Similarly strong Pan-essential genes such as POLD1 have chronos scores <2. Thus the fitness effect of CDK12 loss is relatively modest. I do wonder if it is possible to propagate CDK12 knockout cells albeit more slowly than wildtype cells.

POINT-BY-POINT RESPONSE TO REVIEWER COMMENTS

Reviewer #1 (Remarks to the Author):

The authors addressed most of my comments and significantly improved the manuscript. I understand some of the technical difficulties, but I still have three requirements that need to be addressed before the acceptance for publication:

1) The home-made P-Thr548 CDK12 antibody is the key reagent linking specific changes in phosphorylation of CDK12 to activated RAS/MAPK pathway and thus the antibody needs to be properly characterized. As stated by the authors, the antibody does not recognize endogenous CDK12 phosphorylated on Thr548 and the western blot presented in the response to the comment 5 of the reviewer 2 shows a lot of bands and none of them appears to be sensitive to the CDK12 knockdowns. To rigorously confirm the specificity of the antibody, the authors need to show that the P-Thr548 CDK12 band in overexpressed and immunoprecipitated CDK12 (Fig. 2i, 3e) is indeed sensitive to phosphatase treatment and CDK12 depletion. It should be also directly stated in the result section that P-Thr548 CDK12 antibody does not recognize the endogenous protein, only overexpressed and immunoprecipitated one (as shown in Figs 2i, 3e).

Reply: To address the reviewer's comment, we immunoprecipitated Myc-tagged CDK12 from HEK293 cells stimulated or not with PMA to activate the RAS/MAPK pathway. The immunoprecipitates were treated with phosphatase or not, and assayed for Thr548 phosphorylation using our home-made phosphospecific antibody. As now shown in Fig. 2i, the results confirm that our antibody is indeed phosphospecific. Together with the experiment showing its specificity towards Thr548 (Fig. 2i), these results validate our antibody for its intended use. For clarity, and as suggested by the reviewer, we also modified the Results section of the manuscript to clearly indicate that the antibody is not potent enough to recognize the endogenous protein.

2) I noted that western blots in Figs. 2i, 3e, 3d and in many other figures in the manuscript are disproportionately cropped (please compare to original western blots presented in the source data file). The incorrect manipulation with the original gels distorts meaning of loading controls, effects of treatments on phosphorylation etc. This is not acceptable and needs to be corrected throughout the manuscript.

Reply: We thank the reviewer for noticing these differences. We have verified the entire manuscript and corrected all figures that may have been disproportionately cropped.

3) I appreciate the authors mention that THZ1 is in fact CDK7 inhibitor (as response to my comment no 8). The added references 44 and 45 are, however, misleading (line 301) as only the reference 46 shows that the THZ1 also partially inhibits CDK12 and CDK13. Given that CDK7 inhibition globally downregulates transcription (ref. 46) and the kinase affects activity of many other CDKs (ref. 44, 45), the limitations of this in vivo experiment need to be clearly stated/discussed in the discussion as no selective CDK12 inhibitor was in fact used for in vivo studies (i.e. the observed outcome can be completely independent of the CDK12 function).

Reply: We have made the requested changes and included a short discussion on how the use of THZ1 affects our ability to conclude on the role of CDK12 *in vivo*. We thank the reviewer for these suggestions.

Reviewer #2 (Remarks to the Author):

The authors have address the previous concerns (or opted to not make some changes). One remaining comment that is easily addressed with a simple reference to the literature remains that of Major comment #4. Specifically, the authors argue that as others have already tested dabrafenib plus trametinib, there is little value in comparing their drug combination to the clinical one in a xenograft model. Fair point. However, related to the importance of putting their findings into context with current clinical practice, it would be ever-so-valuable to make a compare the response the authors document in this study with the aforementioned published studies. With the exception of this last easily addressed point, the authors were highly responsive.

Reply: As suggested by the reviewer, the manuscript now compares our xenograft results with the reported effects of a combination of trametinib and dabrafenib. Overall, it appears clear that a combination of trametinib and dabrafenib is more effective at inhibiting A375 xenograft growth than a THZ1/MLN120B combination. Obviously, more work will be required to increase the selectivity, potency and pharmacology of these drugs.

In terms of other previous concerns, I leave it to the authors to decide if they would like to further revise their manuscript in regards to:

Major comment #5- The authors clearly demonstrate that their phospho-specific antibody cannot detect endogenous CDK12 phosphorylation, thank you. Related to this, perhaps the authors could clarify when endogenous versus ectopic proteins are being analyzed in the text. While to the authors' credit, these details can be found in the methods or legends, having it in the text really helps navigate the experiments.

Reply: We made additional mentions in the Results section to indicate when endogenous versus ectopic CDK12 proteins were analyzed.

Minor comment #7- The authors make the case that because ERK is the only substrate for MEK that a gain-of-function mutant of ERK is not required. I note here for the purpose of information that there are both ERK activation mutants and reports of MEK phosphorylating other proteins (although admittedly very few and it is true that the dogma is that MEK only phosphorylates ERK).

Reply: We appreciate the suggestion by the reviewer, but we think that the manuscript sufficiently describes ERK1/2 as upstream regulators of CDK12 phosphorylation. This was done using *in vitro* kinase activity assays with recombinant proteins, the use of multiple pharmacological inhibitors to MEK1/2 and ERK1/2, and the use of a constitutively activated MEK1 mutant.

Minor comment #9- To their credit, the authors generated and then tested a Thr548>Asp mutant of CDK12, finding it did not increase activity. Please consider including this as a supplemental figure, as your hard work could save someone the frustration of trying this experiment. Moreover,

as it is such an obvious epistatic experiment, please consider noting that the mutant does not increase activity in the text.

Reply: Again, we thank the reviewer for this comment. We are always careful when analyzing potential phosphomimetic constructs, as the addition of a negative charge does not always mimic a phosphate group. While negatively charged residues can sometimes induce conformational changes normally triggered by phosphorylation, they are very poor at promoting protein-protein interactions. It is thus difficult to conclude when results are negative, as the amino acid change may not functionally mimic the presence of a phosphate group. Generally, we include positive results using such constructs, but tend to exclude these data when the results are negative because of our inability to provide a useful explanation or conclusion. We hope the reviewer agrees with our view.

Reviewer #3 (Remarks to the Author):

Generally I think the authors have done a good job of responding to my comments. Regarding point 9 relating to DepMap the authors should look at the scores for essential melanoma genes - for example SOX10 or BRAF in BRAF mutant lines. In both cases the genes are strongly essential (with Chronos scores less than -1). Similarly strong Pan-essential genes such as POLD1 have chronos scores <2. Thus the fitness effect of CDK12 loss is relatively modest. I do wonder if it is possible to propagate CDK12 knockout cells albeit more slowly than wildtype cells.

Reply: We have not attempted to generate CDK12 KO cells because DepMap classified it as a common essential gene. We agree that some genes appear to be more essential than others, and that it may be possible to generate CDK12 KO cells based on its DepMap score.

End of point-by-point responses.

Reviewers' Comments:

Reviewer #1:

Remarks to the Author:

My concerns have been addressed and I believe also the remaining questions of the other two reviewers. I recommend publication and congratulate on the manuscript.